# Cholesterol-induced LRP3 downregulation promotes cartilage degeneration in osteoarthritis by targeting Syndecan-4

Chenxi Cao[1,2,3,4], Yuanyuan Shi[1,2,3,4], Xin Zhang[1,2,3], Qi Li[1,2,3], Jiahao Zhang[1,2,3], Fengyuan Zhao[1,2,3], Qingyang Meng[1,2,3], Wenli Dai[1,2,3], Zhenlong Liu[1,2,3], Wenqiang Yan[1,2,3], Xiaoning Duan[1,2,3], Jiying Zhang[1,2,3], Xin Fu[1,2,3], Jin Cheng[1,2,3] ✉, Xiaoqing Hu[1,2,3] ✉ & Yingfang Ao[1,2,3] ✉

Emerging evidence suggests that osteoarthritis is associated with high cholesterol levels in some osteoarthritis patients. However, the specific mechanism under this metabolic osteoarthritis phenotype remains unclear. We find that cholesterol metabolism-related gene, LRP3 (low-density lipoprotein receptor-related protein 3) is significantly reduced in high-cholesterol diet mouse's cartilage. By using *Lrp3*[−/−] mice in vivo and LRP3 lentiviral-transduced chondrocytes in vitro, we identify that LRP3 positively regulate chondrocyte extracellular matrix metabolism, and its deficiency aggravate the degeneration of cartilage. Regardless of diet, LRP3 overexpression in cartilage attenuate anterior cruciate ligament transection induced osteoarthritis progression in rats and *Lrp3* knockout-induced osteoarthritis progression in mice. LRP3 knockdown upregulate syndecan-4 by activating the Ras signaling pathway. We identify syndecan-4 as a downstream molecular target of LRP3 in osteoarthritis pathogenesis. These findings suggest that cholesterol-LRP3-syndecan-4 axis plays critical roles in osteoarthritis development, and LRP3 gene therapy may provide a therapeutic regimen for osteoarthritis treatment.

Osteoarthritis (OA) is the most common form of degenerative arthropathy, with clinical manifestations such as pain, limited mobility, and joint deformities, which seriously affect the patient's ability to work and quality of life[1]. Despite the high prevalence and societal impact, the etiology of OA remains unclear. OA is a multifactorial disease with many possible causes, including aging[2], trauma[3], sex[4], abnormal biomechanics[5], and obesity[6]. Furthermore, new evidence supports a link between OA and metabolic syndromes (MetS), such as hypertension, insulin resistance, and dyslipidemia, which are associated with an increased risk of OA[7]. In the past few years, many animal studies have been conducted to reveal the role of cholesterol in the pathogenesis of OA[8–11]. These findings indicated that different OA animal models fed with a high-cholesterol diet led to increased cartilage damage, elevated adiposity-derived inflammation, and increased OA severity. In the field of clinical research, several lines of evidence have demonstrated that some OA patients have significantly higher serum levels of low-density lipoprotein (LDL) compared to healthy controls[12,13]. Al-Arfaj AS' research showed an association between high serum cholesterol levels and both knee and generalized OA, proving that hypercholesterolemia may be an independent risk factor for systemic OA[14]. Meanwhile, a cross-sectional study in Korea showed that high cholesterol was positively correlated with OA pain[15]. The above evidence indicates it is highly likely that there is a metabolic OA phenotype that is associated with high cholesterol levels.

[1]Department of Sports Medicine, Peking University Third Hospital, 49 North Garden Road, Haidian District, Beijing 100191, People's Republic of China. [2]Institute of Sports Medicine of Peking University, 49 North Garden Road, Haidian District, Beijing 100191, People's Republic of China. [3]Beijing Key Laboratory of Sports Injuries, 49 North Garden Road, Haidian District, Beijing 100191, People's Republic of China. [4]These authors contributed equally: Chenxi Cao, Yuanyuan Shi. ✉e-mail: chengjin@bjmu.edu.cn; huxiaoqingbd01@sina.com; aoyingfang@163.com

In addition, anomalous homeostasis of cholesterol is also shown to be one of the characteristic features of OA development, with an accumulation of cholesterol found particularly in the superficial zone of the cartilage[16]. Functional genomics analysis of osteoarthritic cartilage and chondrocytes added stronger evidence supporting the role of cholesterol-related genes in OA pathogenesis[7]. Wan-Su Choi et al.'s research on *Nature* indicated that OA is a disease associated with metabolic disorders and suggest that targeting the CH25H (cholesterol 25-hydroxylase)-CYP7B1(25-hydroxycholesterol 7α-hydroxylase)-RORα axis of cholesterol metabolism may provide a therapeutic avenue for treating osteoarthritis[10]. Based on existing clinical and basic research, cholesterol metabolism and its related genes may be involved in the occurrence and development of OA, although the relationship between high cholesterol and OA is still controversial in the field of clinical cohort research[17–19]. However, the underlying molecular mechanism of cholesterol metabolism and its related genes involved in OA pathogenesis is still unclear and needs to be explored[20].

The LDL receptor-related protein (LRP) family is a group of evolutionarily conserved cell surface receptors, involved in the regulation of cholesterol metabolism[21], and has other biological functions as well, including cell signal transduction, synaptic plasticity regulation, and cell fate determination[22,23]. In recent years, studies have found that the LRPs are related to the onset and development of OA[24]. LRP5 has been shown to elevate OA cartilage, and its haplotype mutations can lead to an increased risk of knee OA in patients[25]. Consistently, *Lrp5* knockout mice have significantly reduced cartilage degeneration compared with wild-type mice[26]. Similarly, heterozygosity for an inactivating mutation of *Lrp6* can also aggravate the progression of OA in mice[27]. LRP3 is one of the LRP family members and expresses in a wide range of human tissues[28]. In contrast to other members of the LRP family that have been thoroughly studied, the biological function of LRP3 has not been elucidated yet, especially its role in OA.

One of the most important characteristics of OA pathogenesis is cartilage extracellular matrix (ECM) degeneration by matrix-degrading enzymes, including aggrecanases and MMPs[29]. Aggrecanases belong to the ADAMTS family of metalloproteinases, and the main ones that degrade aggrecans are ADAMTS-4 and ADAMTS-5[30]. A recent study showed that syndecan-4 (SDC4) was required for ADAMTS-5 activation in chondrocytes during OA, and played an important role in cartilage ECM degeneration[31]. However, the role of cholesterol metabolism-related gene LRP3 in the degeneration of OA cartilage, the relationship, and the mutual regulation mechanism that exists between LRP3 and SDC4 is currently unclear. Here, we investigated the functional role of the cholesterol–LRP3–SDC4 axis in OA development. The elucidation of such mechanisms could facilitate the development of new and effective therapies for the treatment of OA.

## Results

### High cholesterol aggravates OA cartilage degeneration
Initially, we collected clinical data of 100 OA patients undergoing total knee replacement (TKA) in Peking University Third Hospital and 91 non-OA patients (meniscal injury as control), including blood total cholesterol (TC), LDL, and body mass index (BMI) (Supplementary Table 1). We found that compared with non-OA patients, TC and LDL were significantly higher in OA patients, accompanied by larger BMI and more metabolic diseases, which indicated that there may be a link between hyperlipidemia and the development of OA (Fig. 1a). Subsequently, we investigated the impact of high cholesterol on human cartilage explants in vitro, using exogenous cholesterol and TNF-α stimulation. Safranin-O and toluidine blue staining showed a significant proteoglycans reduction in response to cholesterol and TNF-α, especially under simultaneous stimulation. Meanwhile, immunohistochemistry (IHC) staining showed that the expression of type II collagen was also significantly reduced (Fig. 1b, c).

To further investigate the impact of cholesterol on the development of OA in vivo, we examined whether a high-cholesterol diet (HCD)−compared with a regular diet (RD)−affects ACLT-induced experimental OA in mice (Fig. 1d). We observed that mice fed with an HCD exhibited increased body weight, serum TC and LDL levels. Hot plate tests showed that HCD significantly increased OA-induced pain post-surgery (Fig. 1e). Meanwhile, safranin O-fast green and type II collagen IHC staining showed more severe cartilage degeneration in HCD mice than its RD counterparts 4 weeks after ACLT surgery. The Osteoarthritis Research Society International (OARSI) scores also showed the same trend (Fig. 1f).

Subsequently, RNA sequencing (RNA-seq) analysis indicated that HCD prominently downregulated the chondrocyte's anabolic genes (*Col2a1, Acan*, and *Sox9*), and upregulated the catabolism genes (*Mmp13* and *Adamts5*) (Fig. 1g). Meanwhile, we corroborated the above genes changes in rat chondrocytes under stimulation of cholesterol in vitro (Fig. 1h, i).

### High cholesterol downregulates the expression of LRP3 in cartilage
Furthermore, we comprehensively screened the results of RNA-seq of HCD and RD mice. Among them, we found that the expression of the LRP family, which is closely related to cholesterol metabolism, was changed significantly (Fig. 2a). To verify whether exogenous cholesterol was taken up and absorbed by chondrocytes in vitro, Filipin staining was performed. The results showed that with the increase of exogenous cholesterol concentration, the fluorescence intensity of Filipin staining gradually increased (Fig. 2b). Then, RT-qPCR was performed on the stimulated chondrocytes. We found that the expressions of *Lrp3* and *Lrp4* were decreased compared with the control group, while the expressions of *Lrp5* and *Lrp6* were increased, and *Lrp3* had the most obvious change (Fig. 2c). Further verification results showed that the expression of LRP3 at both the mRNA and the protein level was significantly decreased after exogenous cholesterol stimulation, and the changing trend was more obvious when inflammatory factors and cholesterol acted together (Fig. 2d). The IHC staining also showed decreased expression of LRP3 protein in both HCD mice cartilage and human cartilage explant stimulated with high cholesterol (Fig. 2e, f).

### The expression of LRP3 is decreased during OA development
In order to further clarify the relationship between the expression of LRP3 and OA, we first collected cartilage tissues from five trauma patients (as control) and five OA patients. Safranin-O and toluidine blue staining showed that compared with normal cartilage, the staining intensity of cartilage tissue in OA patients was significantly lower (Fig. 3a, b). IHC staining confirmed that LRP3 protein was downregulated in OA cartilage (Fig. 3c, d). We next investigated the LRP3 expression in the cartilage of various murine OA models. In the ACLT-induced rat OA model, OA was confirmed by safranin O-fast green staining and OARSI score (Fig. 3e, f). Cartilage from ACLT surgery groups exhibited a dramatic reduction in LRP3 expression compared with sham control (Fig. 3g, h). Similar observations were observed in the senescence-induced mouse OA model. The articular cartilage of 12-month-old mice more severely degenerated than that of 3-month-old mice, and the content of proteoglycans was significantly reduced (Fig. 3i, j), while LRP3 protein was also significantly reduced (Fig. 3k, l). Moreover, we found that *Lrp3* and the anabolic genes (*Col2a1, Acan*, and *Sox9*) were prominently downregulated in rat OA chondrocytes induced by IL-1β or TNF-α (Fig. 3m and Supplementary Fig. 1a). Immunofluorescence co-staining and western blot also proved that the protein level of LRP3 was decreased significantly in rat OA chondrocytes model (Fig. 3n and Supplementary Fig. 1b). These results all confirmed that the expression of LRP3 in chondrocytes was decreased during OA development.

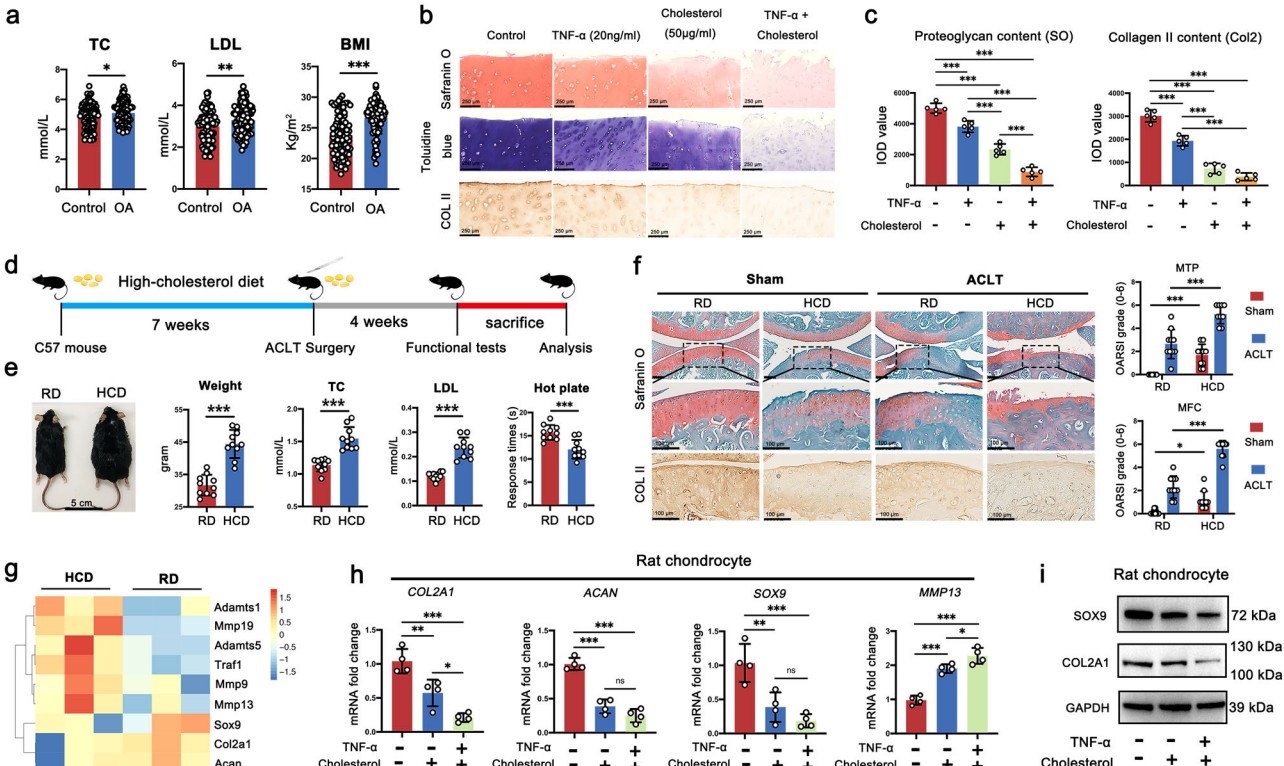

**Fig. 1 | High cholesterol aggravates OA cartilage degeneration. a** Levels of serum total cholesterol (TC), low-density lipoprotein (LDL), and body mass index (BMI) in OA ($n$ = 100) or meniscal injured (as control) patients ($n$ = 91) (unpaired two-tailed Student's $t$-test). **b** Safranin-O, toluidine blue, and type II collagen immunohistochemistry (IHC) staining in human cartilage explants treated with cholesterol (50 μg/ml) and/or TNF-α (20 ng/ml) for 14 d ($n$ = 5). **c** The integrated optical density (IOD) value for proteoglycan content and type II collagen (COLII) determined by safranin O and immunohistochemistry staining ($n$ = 5, one-way ANOVA). **d** Scheme of the high-cholesterol diet (HCD) feeding and ACLT surgery in mice. **e** The general view, body weight, serum TC, serum LDL, and pain response times of HCD mice compared with RD mice ($n$ = 10, two-tailed Student's $t$-test). **f** Representative images of safranin O-fast green, IHC staining of COLII, and the corresponding Osteoarthritis Research Society International (OARSI) scores from HCD mice and RD mice subjected to Sham or ACLT operation for 4 weeks ($n$ = 9, one-way ANOVA). **g** Heatmap of mRNA-seq analysis in cartilage tissue of HCD and RD mice. **h** Quantification of mRNA levels for *COL2A1*, *ACAN*, *SOX9*, and *MMP13* in rat chondrocytes treated with cholesterol (50 μg/ml) and/or TNF-α (20 ng/ml) for 3 d ($n$ = 4, one-way ANOVA). **i** The protein levels of SOX9 and COL2A1 in rat chondrocytes treated with cholesterol (50 μg/ml) and/or TNF-α (20 ng/ml) for 5d ($n$ = 3). Data are shown as the mean ± SD. *$P$ < 0.05; **$P$ < 0.01; ***$P$ < 0.001. $n$ indicates the number of biologically independent samples, mice per group, or human specimens.

## LRP3 positively regulates the metabolism of chondrocytes ECM

To further identify the biological functions of LRP3 in OA, we constructed LRP3 knockdown (Lv-siLrp3) and overexpression (Lv-Lrp3) lentiviral vectors. Through our preliminary experiments, the most suitable MOI for lentivirus infection of rat chondrocytes was 100 (Supplementary Fig. 2a, b). The efficiency of gene knockdown and overexpression was verified by RT-PCR (Supplementary Fig. 2c, d). Then, we found that LRP3 knockdown led to a decrease *Col2a1, Acan*, and *Sox9* mRNA expression in normal rat chondrocytes (Fig. 4a). Western blot analysis also showed a downregulation expression of COL2A1 and SOX9 protein (Fig. 4b). Furthermore, we also found LRP3 knockdown induced the catabolism of chondrocytes ECM. The proteoglycan content was markedly reduced in the three-dimensional pellet-cultured rat chondrocytes (Fig. 4c). The glycosaminoglycan (GAG) content was also significantly reduced in plate-cultured rat chondrocytes testing by alcian blue staining and dimethylmethylene blue (DMMB) assays (Fig. 4d, e). On the contrary, overexpression of LRP3 in TNF-α-induced rat OA chondrocytes significantly increased the expression of COL2A1, ACAN, and SOX9 to near normal levels, both at the mRNA level and the protein level (Fig. 4f, g). Safranin O and toluidine blue staining also confirmed that after overexpression of LRP3, the proteoglycan content of OA chondrocyte pellets was increased significantly (Fig. 4h). In addition, the GAG content of OA chondrocytes caused by inflammatory factors had been restored

(Fig. 4i, j). Those observations revealed that LRP3 positively regulated the metabolism of chondrocytes ECM.

## LRP3 deficiency aggravates the degeneration of knee joint cartilage

We further identified the role of LRP3 in the pathogenesis of OA in vivo using Lv-siLrp3-mediated rats OA model and global LRP3 knockout mice (*Lrp3*[−/−] mice). Intra-articular injection of Lv-siLrp3 resulted in the effective downregulation of LRP3 in cartilage, caused a loss of cartilage GAG and type II collagen, and led to more serious OA performance compared with the vehicle group (Supplementary Fig. 3). Furthermore, the *Lrp3*[−/−] mice were generated and the knockout efficiency was confirmed by PCR screening and IHC staining in various tissues, including cartilage, bone, synovium, and meniscus (Supplementary Fig. 4). We further compared the difference between *Lrp3*[−/−] mice and wild type (WT) mice in physical development. The results showed that there was no obvious disability and dysplasia in *Lrp3*[−/−] mice (Supplementary Fig. 5a). Under regular diet (RD), adult (3 months) *Lrp3*[−/−] and WT mice had no significant difference in body weight (Supplementary Fig. 5b). However, in the case of HCD, the weight of the WT mice was significantly greater than the weight of *Lrp3*[−/−] mice (Supplementary Fig. 5c). Similar to WT mice, *Lrp3*[−/−] mice also gained weight (Supplementary Fig. 5d) and TC in response to HCD (Supplementary Fig. 5e).

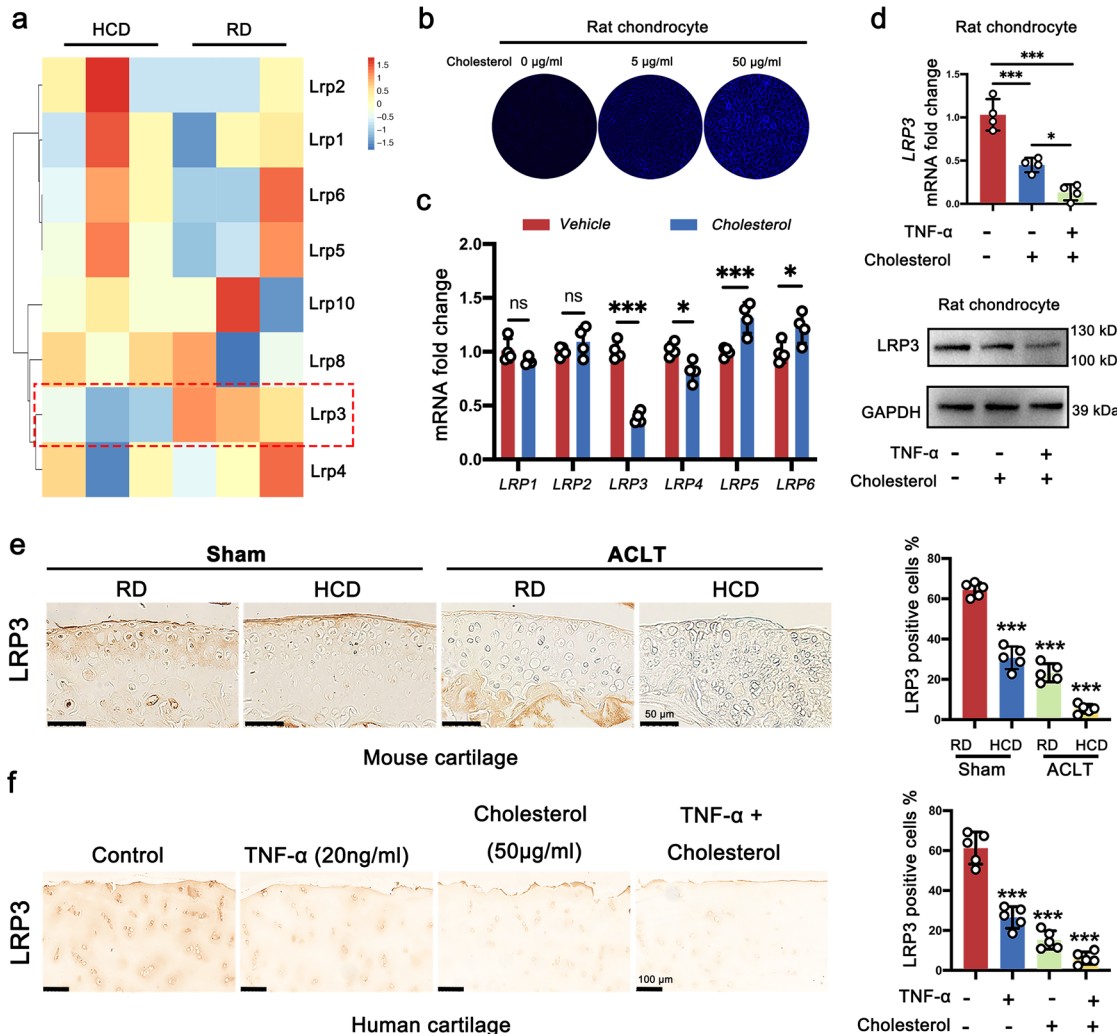

**Fig. 2 | High cholesterol downregulates the expression of LRP3 in cartilage.**
**a** Heatmap of mRNA-seq analysis in cartilage tissue of HCD and RD mice. **b** Filipin staining to detect cholesterol in rat chondrocytes treated with cholesterol (0, 5, and 50 μg/ml). **c** Quantification of mRNA levels for *LRP1*, *LRP2*, *LRP3*, *LRP4*, *LRP5*, and *LRP6* in rat chondrocytes treated with cholesterol (50 μg/ml) (*n* = 4, two-tailed Student's *t*-test). **d** Quantification of LRP3 mRNA (*n* = 4, one-way ANOVA) and protein levels in rat chondrocytes treated with cholesterol (50 μg/ml) and/or TNF-α (20 ng/ml). **e** IHC staining for LRP3 in HCD mice and RD mice subjected to Sham or ACLT operation. Quantification of LRP3-positive cells in mouse knee joint cartilage tissues (*n* = 5, one-way ANOVA). **f** IHC staining for LRP3 in human cartilage explants treated with cholesterol (50 μg/ml) and/or TNF-α (20 ng/ml). Quantification of LRP3-positive cells in human knee joint cartilage tissues (*n* = 5, one-way ANOVA). Data are shown as the mean ± SD. *$P < 0.05$; **$P < 0.01$; ***$P < 0.001$. *n* indicates the number of biologically independent samples, mice per group, or human specimens.

Compared with WT counterparts, more severe ACLT-induced cartilage degeneration was observed in *Lrp3*⁻/⁻ mice at 4 weeks (Supplementary Fig. 6a, b) and 8 weeks (Fig. 4k, l), either in the sham or surgery group. Type II collagen was also decreased in the cartilage of *Lrp3*⁻/⁻ mice (Fig. 4k, m). Micro-CT 3D reconstruction images revealed that the ectopic osteophytes of *Lrp3*⁻/⁻ mice were significantly increased, the joints surface were obviously rough and the joint space was reduced (Fig. 4n and Supplementary Fig. 6c). In addition, the hot plate test showed that the pain response of *Lrp3*⁻/⁻ mice was more sensitive than WT mice after surgery (Fig. 4o). What's more, we further compared the OA phenotype between the 12-month-old naturally senile *Lrp3*⁻/⁻ mice and WT mice. Safranin O-fast green staining showed that proteoglycan was barely expressed in the cartilage of aging *Lrp3*⁻/⁻ mice and OARSI scores exhibited more serious OA progression (Fig. 4p). IHC analysis also showed that type II collagen was significantly decreased in *Lrp3*⁻/⁻ mice cartilage (Fig. 4q). Collectively, the above results revealed that deficiency of LRP3 accelerated both post-injury OA and senescence-induced OA development.

**Overexpression of LRP3 in cartilage attenuates OA progression**
Next, we sought to investigate whether the overexpression of LRP3 could attenuate the progression of OA in vivo. According to the research design, 50 μl Lv-Lrp3 (low dose: $1 \times 10^5$ lentiviral particles; high dose: $1 \times 10^6$ lentiviral particles) or negative control lentiviral (Lv-con335) was injected into the knee joints once a week for 6 weeks after ACLT (Fig. 5a). Efficiency of lentiviral infection in the knee cartilage was assessed by LRP3 IHC staining and RT-PCR. Assessing by safranin O-fast green and IHC staining, we observed that LRP3 overexpression markedly inhibited articular cartilage erosion, rescued the proteoglycan and type II collagen relative to vehicle-treated ACLT controls (Fig. 5b, c). The OARSI scores also suggested Lv-Lrp3-treated knee joints exhibited milder OA phenotype (Fig. 5d). Functional tests were applied for assessing the OA alleviation effect after Lv-Lrp3 treatment. The hot plate and weight-bearing tests showed that intra-articular injection of Lv-Lrp3 significantly reduced rat's OA-induced pain (Fig. 5e). Furthermore, the RT-PCR showed that the anabolic genes (*Col2a1*, *Acan*, and *Sox9*) were upregulated and the catabolic genes (*Adamts5* and *Mmp13*) were down-regulated after the

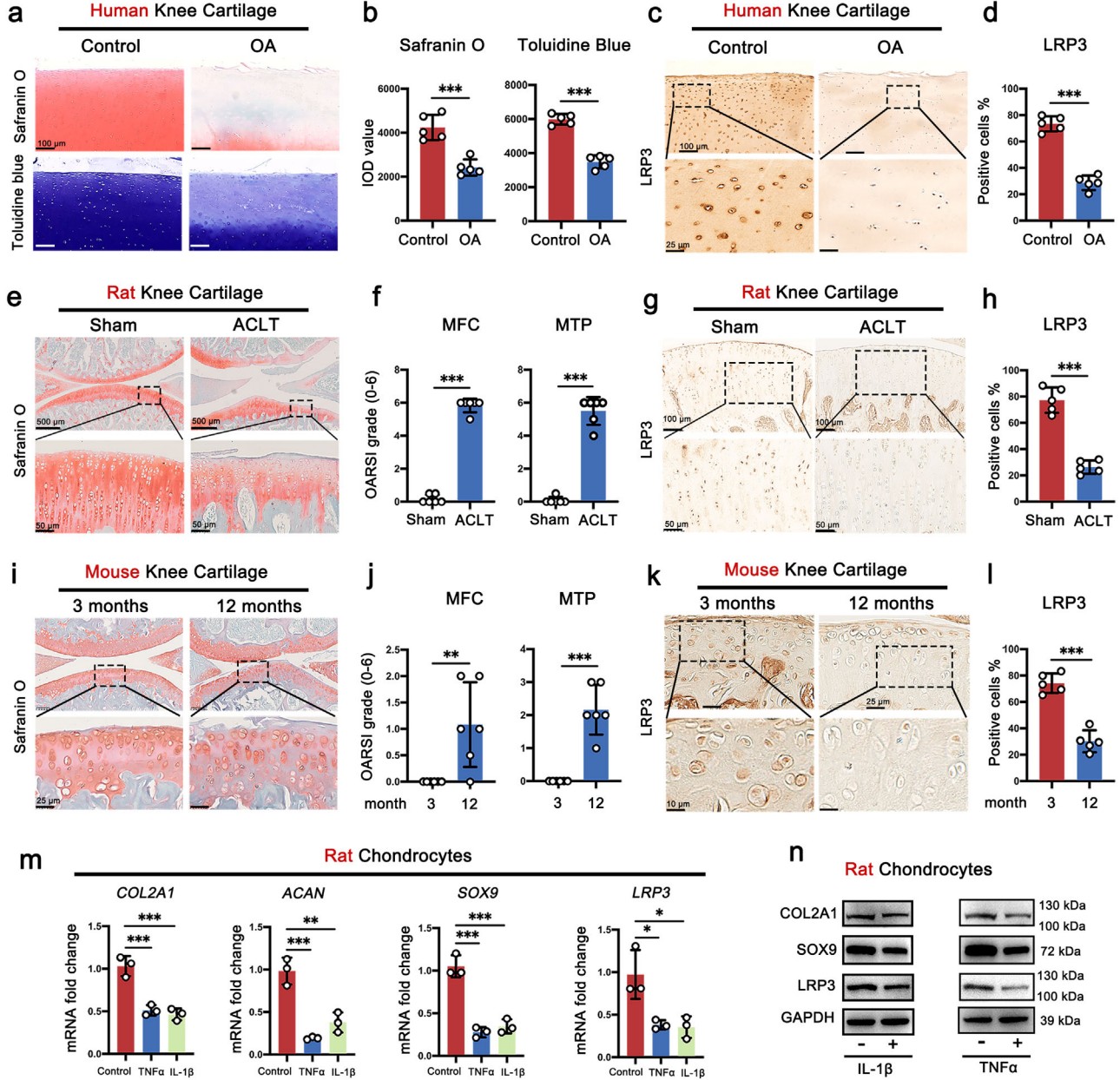

**Fig. 3 | The expression of LRP3 is decreased during OA development. a** Safranin-O and toluidine blue staining in human OA and normal cartilage ($n = 5$). **b** The IOD value for human cartilage proteoglycan content determined by safranin O and toluidine blue staining ($n = 5$, two-tailed Student's $t$-test). **c** IHC staining for LRP3 in human OA and normal cartilage ($n = 5$). **d** Quantification of LRP3-positive cells in human OA and normal cartilage ($n = 5$, two-tailed Student's $t$-test). **e** Representative images of safranin O-fast green staining in rat Sham and ACLT knee cartilage ($n = 5$). **f** OARSI scores of rat Sham and ACLT knee cartilage ($n = 5$, two-tailed Student's $t$-test). **g** IHC staining for LRP3 in rat Sham and ACLT tibial cartilage ($n = 5$). **h** Quantification of LRP3-positive cells in rat Sham and ACLT tibial cartilage ($n = 5$, two-tailed Student's $t$-test). **i** Representative images of safranin O-fast green staining in 3 months and 12 months mouse knee cartilage ($n = 5$). **j** OARSI scores of 3 months and 12 months mouse knee cartilage ($n = 5$, two-tailed Student's $t$-test). **k** IHC staining for LRP3 in 3 months and 12 months mouse tibial cartilage ($n = 5$). **l** Quantification of LRP3-positive cells in 3 months and 12 months mouse tibial cartilage ($n = 5$, two-tailed Student's $t$-test). **m** Quantification of mRNA levels for *COL2A1*, *ACAN*, *SOX9*, and *LRP3* in rat chondrocytes treated with IL-1β (10 ng/ml) and TNF-α (20 ng/ml) for 3 d ($n = 3$, one-way ANOVA). **n** The protein levels of SOX9, COL2A1, and LRP3 in rat chondrocytes treated with IL-1β (10 ng/ml) and TNF-α (20 ng/ml) for 5 d ($n = 3$). Data are shown as the mean ± SD. *$P < 0.05$; **$P < 0.01$; ***$P < 0.001$. $n$ indicates the number of biologically independent samples, mice per group or human specimens.

Lv-Lrp3 injection treatment (Fig. 5f), which was consistent with histological analysis. Then, we used micro-CT to examine the effect of Lv-Lrp3 intra-articular injection on osteophyte formation and subchondral bone remodeling in ACLT rats. The CT images showed that the articular surface of the Lv-Lrp3 injection treatment group was smoother than that of the vehicle treatment group, and the generation of ectopic osteophytes was reduced. Besides, Lv-Lrp3 injection treatment increased the tibial trabecular bone volume per total volume (BV/TV) and decreased the trabecular bone pattern factor (Tb. Pf) compared with vehicle group post ACLT (Fig. 5g). Meanwhile, a nanoindentation test was carried out to determine the biomechanical properties of the rat's articular cartilage. The Lv-Lrp3 injection treatment group exhibited a higher elastic modulus and hardness compared with those in the vehicle-treated group, which were closer to normal cartilage tissue. And the load–depth curve revealed consistent results (Fig. 5h).

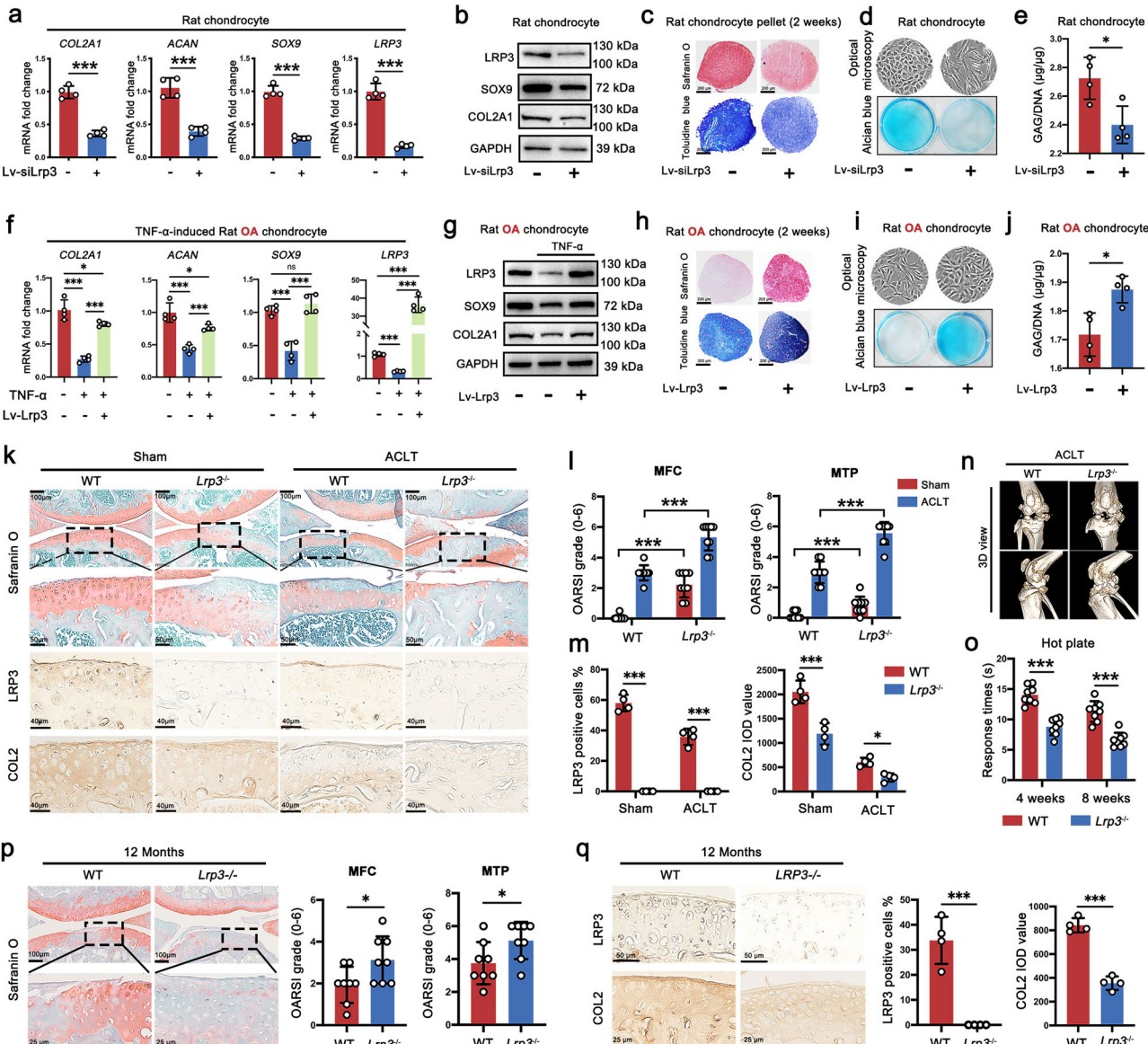

**Fig. 4 | LRP3 positively regulates the metabolism of chondrocytes ECM, and the deficiency of LRP3 aggravates the degeneration of knee joint cartilage.**
**a** Quantification of mRNA ($n = 4$, two-tailed Student's $t$-test) and **b** proteins levels ($n = 3$) in rat chondrocytes treated with Lv-con313 ($n = 4$, two-tailed Student's $t$-test). **c** Safranin O and toluidine blue staining of rat chondrocyte pellets treated with Lv-con313 ($n = 3$). **d** Optical microscopy, alcian blue staining, and **e** DMMB assays to examine the glycosaminoglycan expression ($n = 4$, two-tailed Student's $t$-test). **f** Quantification of mRNA ($n = 4$, one-way ANOVA) and **g** proteins levels ($n = 3$) in TNF-α (20 ng/ml)-induced rat OA chondrocytes treated with Lv-con335. **h** Safranin O and toluidine blue staining of TNF-α (20 ng/ml)-induced rat OA chondrocyte pellets treated with Lv-con313 ($n = 3$). **i** Optical microscopy, alcian blue staining, and **j** DMMB assays to examine the glycosaminoglycan expression ($n = 4$, two-tailed Student's $t$-test). **k** Representative images of safranin O-fast green, LRP3, and COLII

IHC staining of knee joints from WT and $Lrp3^{-/-}$ mice subjected to Sham or ACLT operation for 8 weeks ($n = 9$). **l** OARSI scores of WT and $Lrp3^{-/-}$ mice subjected to Sham or ACLT operation for 8 weeks ($n = 9$, one-way ANOVA). **m** Quantification of LRP3-positive cells and IOD value for COLII ($n = 4$, two-tailed Student's $t$-test). **n** Representative 3D view images of micro-CT of WT mice and $Lrp3^{-/-}$ mice subjected to ACLT operation for 8 weeks ($n = 4$). **o** Pain response times of WT or $Lrp3^{-/-}$ mice at 4 w and 8 w post-surgery ($n = 8$, two-tailed Student's $t$-test). **p** Representative images of safranin O-fast green and OARSI scores of 12 months WT mice and $Lrp3^{-/-}$ mice ($n = 8$, two-tailed Student's $t$-test). **q** LRP3 and COLII IHC staining of 12 months WT mice and $Lrp3^{-/-}$ mice ($n = 8$, two-tailed Student's $t$-test). Data are shown as the mean ± SD. *$P < 0.05$; **$P < 0.01$; ***$P < 0.001$. $n$ indicates the number of biologically independent samples or mice per group.

Moreover, we performed rescue experiments on $Lrp3^{-/-}$ mice after ACLT with Lv-Lrp3 (Supplementary Fig. 7a). Efficiency of LRP3 rescue was assessed by LRP3 immunohistochemical staining. The results showed that after 4 weeks of Lv-Lrp3 injection, the OA phenotypes of $Lrp3^{-/-}$ mice were significantly attenuated and rescued (Supplementary Fig. 7b). The OARSI scores were significantly reduced, and the expression of type II collagen was significantly increased (Supplementary Fig. 7c, d). Therefore, we concluded that overexpression of LRP3 attenuated cartilage degradation under OA conditions.

## Knockdown of LRP3 upregulates the expression of SDC4 by activating the Ras/Raf/MEK/ERK signaling pathway in chondrocytes

To identify the underlying mechanisms of LRP3 on OA development, we performed RNA-seq analyses in the samples of rat chondrocytes treated with Lv-siLrp3 or Lv-con313. In all, 5903 genes were differentially expressed after $Lrp3$ knockdown. Among them, transcripts of 3116 genes were upregulated, whereas 2787 genes were downregulated (Fig. 6a, b). After careful screening, we found that SDC4,

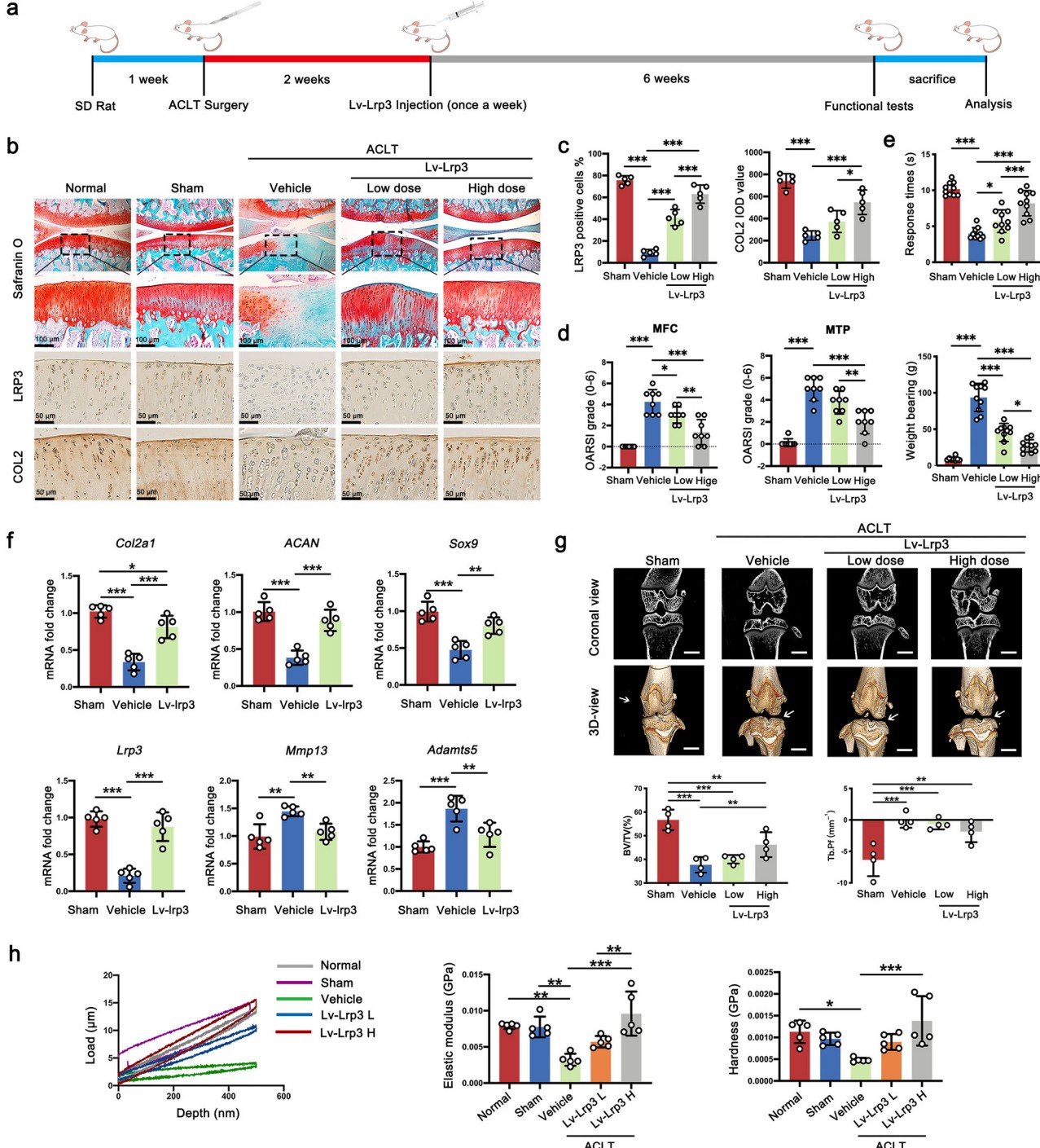

**Fig. 5 | Overexpression of LRP3 in cartilage attenuates OA progression.**
**a** Scheme of the OA treatment with intra-articular injection of Lv-Lrp3 in rats.
**b** Representative images of safranin O-fast green, COLII, and LRP3 IHC staining of Normal, Sham, Vehicle, and Lv-Lrp3 treatment rats at 6 w, insets indicate the regions shown in the enlarged images (*n* = 8). **c** Quantification of LRP3-positive cells and IOD value for COLII in Normal, Sham, Vehicle, and Lv-Lrp3 treatment rats (*n* = 5, one-way ANOVA). **d** The corresponding OARSI scores of Normal, Sham, Vehicle, and Lv-Lrp3 treatment rats at 6 w (*n* = 8, one-way ANOVA). **e** The pain response of rats was analyzed using a hot plate and weight-bearing tests (*n* = 10).
**f** Quantification of mRNA levels for *Col2a1*, *ACAN*, *Sox9*, *Lrp3*, *Mmp13*, and *Adamts5*

in articular cartilage obtained from Sham, Vehicle, and Lv-Lrp3 (high dose group) treatment rats (*n* = 5, one-way ANOVA). **g** Micro-CT Coronal and 3D view and the quantitative micro-CT analysis of tibial subchondral bone with trabecular bone volume per total volume (BV/TV) and trabecular bone pattern factor (Tb. Pf) in Sham, Vehicle, and Lv-Lrp3 (high dose group) treatment rats (*n* = 4, one-way ANOVA). **h** Biomechanical properties of cartilage samples from Normal, Sham, Vehicle and Lv-Lrp3 treatment rats at 6 w were assessed using a nanoindentation test (*n* = 5, one-way ANOVA). Data are shown as the mean ± SD. *P* < 0.05; **P* < 0.01; ***P* < 0.001. *n* indicates the number of biologically independent samples or mice per group.

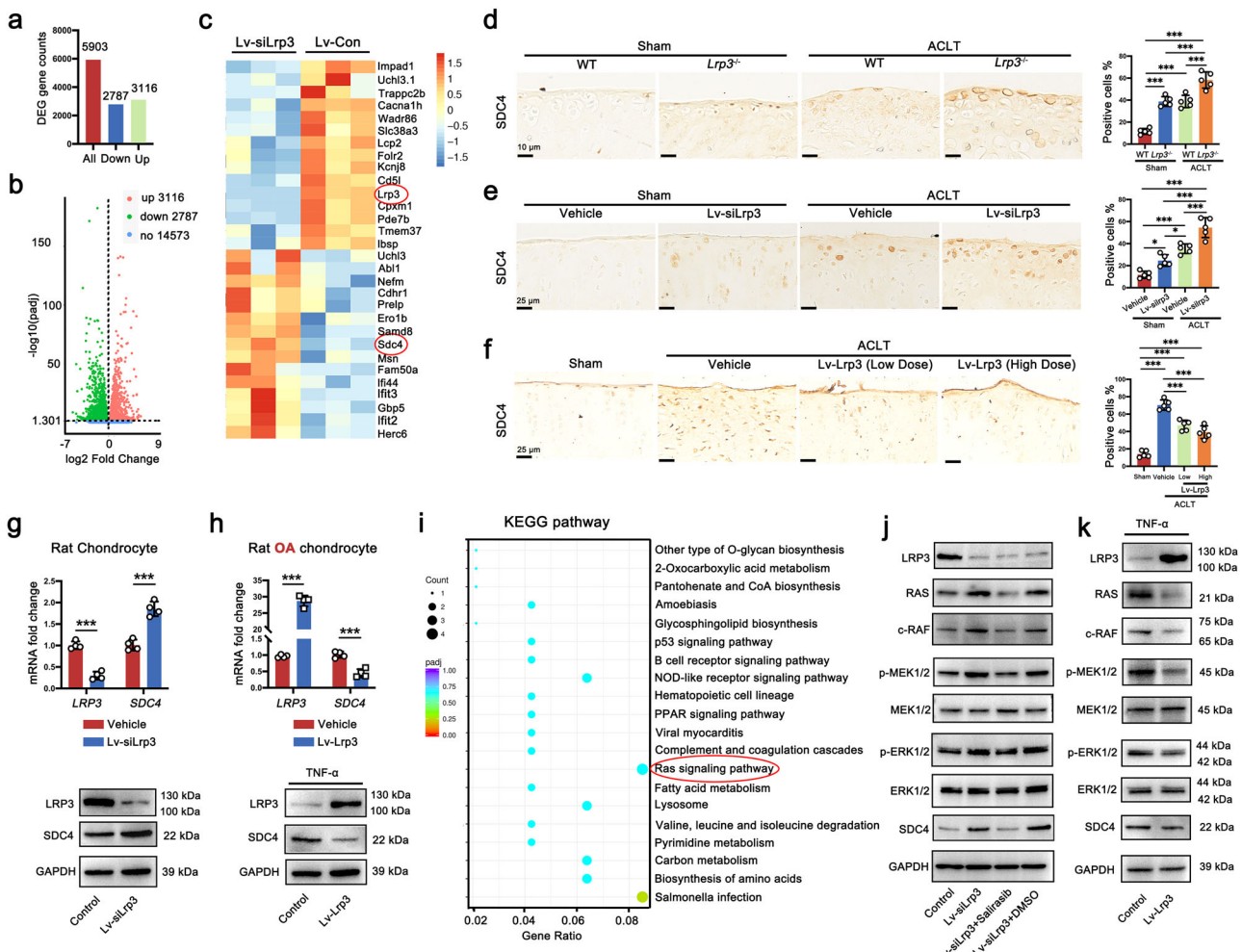

**Fig. 6 | Knockdown of LRP3 upregulates the expression of SDC4 by activating the Ras/Raf/MEK/ERK signaling pathway in chondrocytes. a** RNA-seq comparison revealed a total of 5930 genes expressed, of which 3116 genes were upregulated and 2787 genes were downregulated. **b** The volcano plot illustrating differential genes from RNA-seq analysis. **c** Heatmap of mRNA-seq analysis in rat chondrocytes treated with Lv-con 313 or Lv-siLrp3. **d** Representative images of LRP3 IHC staining of knee joints from WT and LRP3$^{-/-}$ mice subjected to Sham or ACLT operation ($n = 5$, one-way ANOVA). **e** Representative images of LRP3 IHC staining of knee joints from vehicle or Lv-siLrp3 injected rats subjected to Sham or ACLT operation ($n = 5$, one-way ANOVA). **f** Representative images of LRP3 IHC staining of knee joints from Sham, Vehicle, and Lv-Lrp3 treatment rats ($n = 5$, one-way ANOVA). **g** Quantification of LRP3 and SDC4 mRNA ($n = 4$, two-tailed Student's $t$-test) and

proteins levels ($n = 3$) in rat chondrocytes treated with Lv-con313 or Lv-siLrp3. **h** Quantification of LRP3 and SDC4 mRNA ($n = 4$, two-tailed Student's $t$-test) and proteins levels ($n = 3$) in TNF-α (20 ng/ml)-induced rat OA chondrocytes treated with Lv-con335 or Lv-Lrp3 ($n = 5$). **i** KEGG pathway analysis of upregulated targets in LRP3-knockdown transcriptome. **j** Western blot analysis of the Ras/Raf/MEK/ERK signaling pathway and SDC4 in rat chondrocytes treated with Lv-con313, Lv-siLrp3, and/or Salirasib ($n = 3$). **k** Western blot analysis of the Ras/Raf/MEK/ERK signaling pathway and SDC4 in TNF-α (20 ng/ml)-induced rat OA chondrocytes treated with Lv-con335 or Lv-Lrp3 ($n = 3$). Data are shown as the mean ± SD. *$P < 0.05$; **$P < 0.01$; ***$P < 0.001$. $n$ indicates the number of biologically independent samples or mice per group.

which is closely related to OA cartilage degeneration, was significantly up-regulated in response to *Lrp3* knockdown (Fig. 6c). It suggested that SDC4 may be the downstream molecular target of LRP3 in regulating the degeneration process of OA cartilage. Thus, we verified the relationship between LRP3 and SDC4. The IHC staining results showed that the expression of SDC4 was significantly higher in the articular cartilage of *Lrp3*$^{-/-}$ mice than that of wild-type mice (Fig. 6d). Similar results were shown in the cartilage tissue of rats intra-articular injected with Lv-siLrp3 (Fig. 6e). Interestingly, we also found that the ACLT-induced upregulation of SDC4 was suppressed significantly after Lv-Lrp3 injection treatment (Fig. 6f). Meanwhile, we used RT-PCR and western blot to confirm that knocking down LRP3 upregulated the expression of SDC4 in normal rat chondrocytes (Fig. 6g), while over-expression of LRP3 effectively downregulated the expression of SDC4 in TNF-α-induced rat OA chondrocytes (Fig. 6h). Therefore, we focused on SDC4 in the subsequent experiments.

Moreover, the Kyoto encyclopedia of genes and genomes (KEGG) pathway analysis revealed that the rat sarcoma (Ras)/rapidly accelerated fibrosarcoma (Raf)/mitogen-activated protein kinase kinase (MEK)/extracellular regulated protein kinases (ERK) signaling pathway was obviously activated after knocking down LRP3 (Fig. 6i). Previous research found that the Ras/Raf/MEK/ERK signaling pathway participated in the regulation of inflammatory response, leading to OA cartilage degradation[32]. At the same time, studies also showed that the phosphorylation of ERK1/2 was inhibited in chondrocytes of *Sdc4*$^{-/-}$ mice[31]. Therefore, we proposed the hypothesis that knocking down LRP3 caused the up-regulation of SDC4 by activating the Ras/Raf/MEK/ERK signaling pathway. Then we performed a western blot to verify the hypothesis. The results showed that knocking down LRP3 significantly increased the expression of Ras, Raf, and the phosphorylation level of MEK1/2 and ERK1/2, while the SDC4 was also significantly increased simultaneously. Meanwhile, the Ras signaling pathway inhibitor

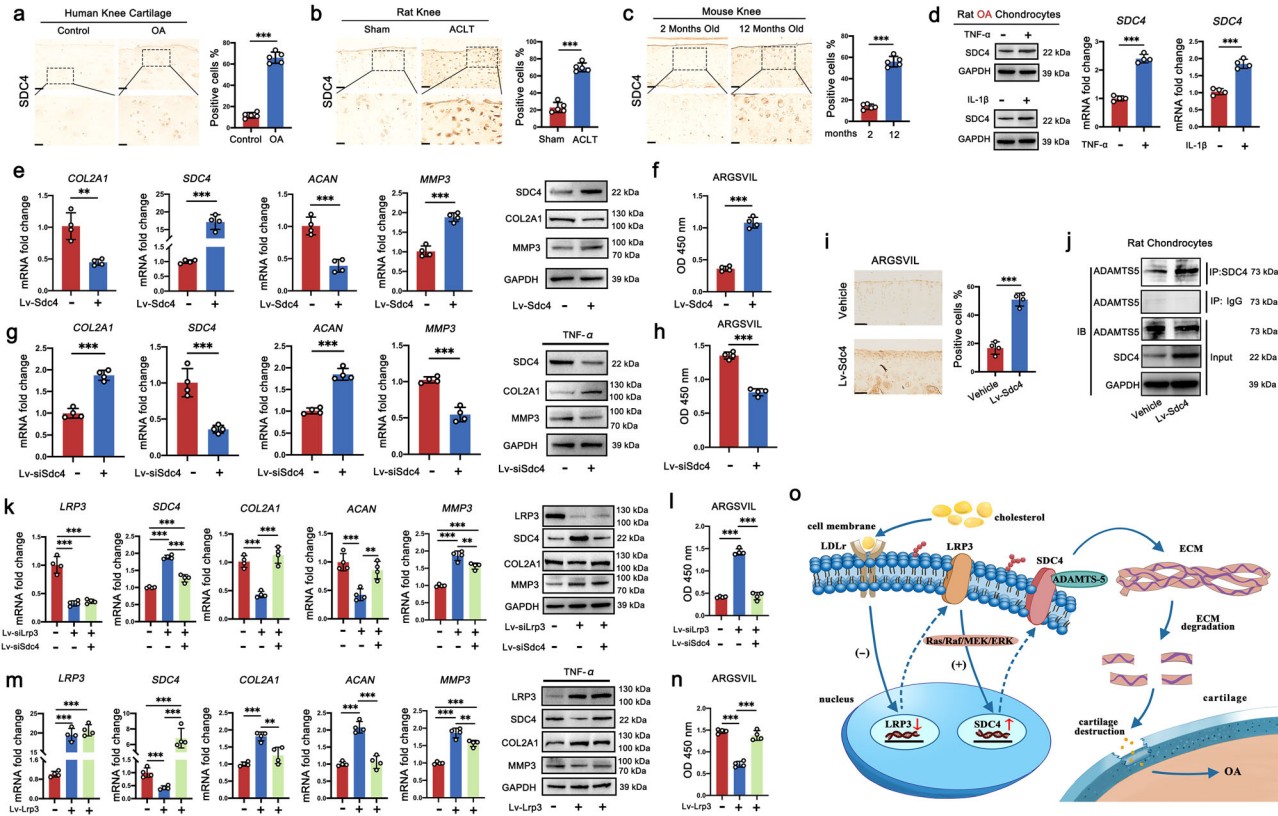

**Fig. 7 | Identification of SDC4 as the downstream molecular target of LRP3 in chondrocytes. a** IHC staining for SDC4 in human cartilage (*n* = 5). **b** IHC staining for SDC4 in rat tibial cartilage (*n* = 5). **c** IHC staining for SDC4 in mouse tibial cartilage, (*n* = 5). **d** Quantification of SDC4 mRNA (*n* = 4, two-tailed Student's *t*-test) and proteins levels (*n* = 3) in rat chondrocytes treated with IL-1β (10 ng/ml) and TNF-α (20 ng/ml). **e** Quantification of mRNA (*n* = 4) and proteins levels (*n* = 3) in rat chondrocytes treated with Lv-Sdc4. **f** The Elisa analysis of ARGSVIL in the cell culture supernatant of rat chondrocytes treated with Lv-con 335 or Lv-Sdc4 (*n* = 4). **g** Quantification of mRNA (*n* = 4) and proteins levels (*n* = 3) in TNF-α (20 ng/ml)-induced rat OA chondrocytes treated with Lv-siSdc4. **h** The Elisa analysis of ARGSVIL in the cell culture supernatant of TNF-α (20 ng/ml)-induced rat OA chondrocytes treated with Lv-siSdc4 (*n* = 4). **i** IHC staining for ARGSVIL in tibial cartilage of vehicle or Lv-Sdc4 injected rats (*n* = 4). **j** The interaction of SDC4 and

ADAMTS5 was verified by co-immunoprecipitation (co-IP) analysis (*n* = 3). **k** Quantification of mRNA (*n* = 4) and proteins levels (*n* = 3) in rat chondrocytes treated with Lv-siLrp3 and Lv-siSdc4. **l** The Elisa analysis of ARGSVIL in the cell culture supernatant of rat chondrocytes treated with Lv-siLrp3 and Lv-siSdc4 (*n* = 4). **m** Quantification of mRNA (*n* = 4) and proteins levels (*n* = 3) in TNF-α (20 ng/ml)-induced rat OA chondrocytes treated with Lv-Lrp3 and Lv-Sdc4. **n** The Elisa analysis of ARGSVIL in the cell culture supernatant of TNF-α (20 ng/ml)-induced rat OA chondrocytes treated with Lv-Lrp3 and Lv-Sdc4 (*n* = 4). **o** Schematic of the role of cholesterol-LRP3-SDC4 axis in the pathogenesis of OA cartilage degeneration. Data are shown as the mean ± SD. \**P* < 0.05; \*\**P* < 0.01; \*\*\**P* < 0.001. For statistical analysis, **a**–**i** were performed by two-tailed Student's *t*-test; **k**–**n** were performed by one-way ANOVA. *n* indicates the number of biologically independent samples, mice per group, or human specimens.

(Salirasib) abolished the upregulation of SDC4 induced by knocking down LRP3 (Fig. 6j). Furthermore, the Ras signaling pathway was obviously activated in the TNF-α-induced rat OA chondrocytes, while overexpression of LRP3 inhibited the activation of Ras signaling pathway and the expression of SDC4 simultaneously (Fig. 6k). In summary, we concluded that LRP3 negatively regulated the expression of SDC4 in chondrocytes through the Ras/Raf/MEK/ERK signaling pathway.

### Identification of SDC4 as the downstream molecular target of LRP3 in chondrocytes

Firstly, we investigated the relationship between the expression of SDC4 and OA. The IHC analysis showed that either in the cartilage of OA patients, ACLT-induced post-traumatic rat OA model, or senescence-induced mouse OA model, the expression of SDC4 was higher than that of their control group respectively (Fig. 7a–c). RT-PCR and western blot demonstrated that SDC4 was also up-regulated in TNF-α and IL-1β induced rat OA chondrocytes (Fig. 7d). Those results all confirmed that the expression of SDC4 was increased during OA progress, which was consistent with the previous studies[33]. We next assessed whether SDC4 had a vital role in cartilage ECM degeneration

in vitro and in vivo. We found that when SDC4 was overexpressed in normal rat chondrocytes by infection of Lv-Sdc4, the expression of COL2A1 and ACAN were decreased significantly, the expression of MMP3 was increased, both in mRNA and protein levels (Fig. 7e). Simultaneously, to determine if SDC4 promoted ECM degeneration, we measured generation of aggrecan fragments using neoepitope recognizing anti-ARGSVIL antibodies. A significant increase in aggrecan neoepitope generation was detected by Elisa assay in the culture supernatant of normal rat chondrocytes by infection of Lv-Sdc4 compared with Lv-con335 (Fig. 7f). Furthermore, knocking down SDC4 in TNF-α-induced OA rat chondrocytes, resulted in an increase of COL2A1 and ACAN and decrease of MMP3 and aggrecan fragmentation (Fig. 7g, h). Then, to evaluate the SDC4 biological function of cartilage degeneration in vivo, Lv-Sdc4 or negative control lentiviral was injected into the knee joints of ACLT-induced rat OA model once a week for 4 weeks (Supplementary Fig. 8a). The Lv-Sdc4 injection group showed a more severe OA phenotype, including the loss of proteoglycans and the increase in OARSI scores (Supplementary Fig. 8b, c). The IHC staining results also showed a decrease of type II collagen, accompanied by the increase of ARGSVIL (Supplementary Fig. 8b, d and Fig. 7i). Therefore, we concluded that overexpression of SDC4 could

aggravate cartilage degradation by breaking down the chondrocyte ECM. Subsequently, we determined whether SDC4 interacted with ADAMTS-5 using co-immunoprecipitation assays. The results showed that after overexpressing of SDC4, pulling down of SDC4 resulted in co-precipitation of more ADAMTS-5 (Fig. 7j). Thus, these data indicated that SDC4 interacted with ADAMTS-5 and MMP3 in the progress of the chondrocyte ECM degeneration.

To further explore whether LRP3 regulates the metabolism of cartilage ECM by targeting SDC4, we knocked down LRP3 and SDC4 in normal rat chondrocytes at the same time. The results showed that knocking down SDC4 alleviated the degradation of chondrocyte ECM caused by the transfection of Lv-siLrp3 (Fig. 7k, l). Meanwhile, SDC4 overexpression counteracted ECM anabolism induced by upregulation of LRP3 in TNF-α-induced rat OA chondrocytes (Fig. 7m, n). In light of these data, we concluded that LRP3 regulates the metabolism of cartilage ECM by targeting SDC4.

## Discussion

In this paper, we have made several key discoveries: (1) high cholesterol stimulation on chondrocytes caused the disorder of cartilage ECM metabolism in vitro and in vivo, leading to cartilage degradation and subsequent OA progression. During this process, the expression of the cholesterol metabolism-related gene, LRP3, was significantly reduced in chondrocytes; (2) the expression of LRP3 was downregulated in degenerated human OA cartilage and several murine models of OA. *Lrp3* gene deletion aggravated cartilage degeneration in mice caused by ACLT, while overexpression of LRP3 in the joint by the lentivirus significantly suppressed cartilage ECM degeneration, thereby inhibiting the progression of OA; (3) knocking down of LRP3 upregulated the expression of SDC4 by activating the Ras/Raf/MEK/ERK signaling pathway in chondrocytes, and overexpression of SDC4 induced cartilage ECM degeneration through binding and activating ADAMTS-5. Taken together, the above observations suggested that the cholesterol−LRP3−SDC4 axis plays an important role in the maintenance of cartilage homeostasis and might be a therapeutic target for the treatment of OA (Fig. 7o).

OA is the most prevalent chronic joint disease that mainly affects individuals over the age of 65[34]. Although several risk factors related to OA have been pointed out, including aging, gender, muscle strength, obesity, and mechanical alignment[1], the pathogenesis of OA remains largely unclear. MetS is a complex group of metabolic disorder syndromes, including a number of conditions, like abdominal obesity, impaired fasting glucose, hypertension (HTN), and dyslipidemia (high cholesterol and low high-density lipoprotein)[35]. Since both MetS and OA pose significant challenges to public health, a large number of studies have been conducted to investigate the association between them. More specifically, Vasiliki Gkretsi et al. reported a positive correlation between increased cholesterol levels and knee, hip, hand, and/or generalized OA through epidemiological studies[36]. A cohort study on knee OA showed that hyperlipidemia is associated with joint pain and an increased prevalence of OA[37]. The pain and OA severity scores of OA patients with hypercholesterolemia were also higher than those of the control group[38]. In addition, Kellgren el al. first showed an association between elevated serum cholesterol levels and hand OA in women[39]. At the same time, a study involving 1003 middle-aged women revealed that elevated serum TC levels were associated with unilateral and bilateral knee OA[40]. In this study, we also found that serum TC and LDL in some OA patients were significantly higher than that of non-OA patients, which verified the positive correlation between clinical hyperlipidemia and OA. As for basic research, both global apolipoprotein E knockout (ApoE$^{−/−}$) mice and rats fed with HCD were proved to have the severity of OA phenotype[10,41], which was consistent with our study. In addition, we used exogenous cholesterol to stimulate human cartilage explants in vitro, and the results showed significant proteoglycans and type II collagen reduction. Similarly,

Saba Farnaghi et al. also found the same change trend when stimulating bovine cartilage explants with cholesterol[41]. Furthermore, studies showed that the administration of atorvastatin can reduce the serum cholesterol level while effectively delaying the progression of OA[9]. Although the above results showed a positive correlation between cholesterol levels and OA, a number of studies have reported contradictory results. The data of Schwager et al.'s research revealed that neither TC, LDL, nor HDL showed a significant association with radiographic or symptomatic OA, as well as cartilage loss, worsening synovitis, or worsening knee pain[17]. In addition, in the Rotterdam study, total/HDL cholesterol ratio was not associated with overweight and hand OA in 3585 participants over 55 years[19]. The differences in study results may be caused by differences in the number of people enrolled in the study cohort, the diagnostic criteria for OA, or the geographical distribution of the population. Although the relationship between high cholesterol and the occurrence and development of OA is still controversial in the clinical cohort study, based on existing clinical and basic research, it is highly likely that there is a metabolic OA phenotype that is associated with high cholesterol levels. However, the mechanisms underlying the complex interactions between lipid metabolism, especially cholesterol metabolism, and OA has not yet been completely elucidated[42].

Cholesterol is an important composition ingredient of cartilage cell membrane lipid dual molecular layer, participating in cartilage cell membrane steady-state maintenance, cell signal transduction, immune response activation, and material transport, and the normal metabolism of cholesterol plays an important role in maintaining cartilage ECM metabolic balance[16,43,44]. However, when an abnormal cholesterol metabolism occurs, a series of pathophysiological changes in cartilage cells can be induced, thereby promoting the development of OA. A previous study found that abnormal cholesterol metabolism can lead to cholesterol accumulation in chondrocytes, causing damage to the structure and function of mitochondria, activating the MAPK inflammatory signaling pathway, leading to irreversible damage to chondrocytes, and causing cartilage degeneration[41]. Excess cholesterol can also be swallowed by synovial macrophages in the joint chamber, resulting in the accumulation of OX-LDL in macrophages, causing ectopic bone formation and aggravating OA progression[45].

With the continuous development and in-depth exploration of OA cartilage and chondrocyte genomics and proteomics, more researchers added stronger evidence supporting the role of cholesterol-related genes in OA pathogenesis. Ashwell et al. studied the transcriptomics characteristics of OA chondrocytes and normal chondrocytes and found that the 30 genes differentially expressed between them were related to matrix molecules, bone development, cell proliferation, and lipid metabolism[46]. Wan-Su Choi et al.'s research found that cholesterol hydroxylase CH25H and CYP7B1 were significantly up-regulated in OA chondrocytes, which mediated ECM degradation by promoting the expression of cartilage ECM degrading enzymes MMP-3, MMP-12, and ADAMTS-5[10]. Our research found that the cholesterol-related genes family, LRPs, have changed significantly. Through RT-qPCR detection of chondrocytes after cholesterol intervention, we found that the expression of *Lrp4* decreased compared with the control group, while the expression of *Lrp5* and *Lrp6* increased, which is consistent with previous studies[24]. Among them, the most significant expression change is *Lrp3*, which has not been reported in previous studies. Further experimental verification results showed that the expression of LRP3 was significantly reduced after exogenous cholesterol stimulation, and the changing trend is more obvious when inflammatory factors and cholesterol work together. This indicates that LRP3 may be a target in OA cartilage degeneration.

As a member of the LRP family, LRP3 was first reported by Hirofumi Ishii et al.[28]. The human *LRP3* gene was mapped to chromosome bands 19q12-q13.2 and detected in a wide range of human tissues, with the highest expression in skeletal muscle and ovary. In addition to its

involvement in lipid metabolism, research on LRP3 is very limited, and its biological function is still unclear[47]. Only one research showed that LRP3 may be the target of microRNA-4739 in the regulation of osteogenic and adipocytic differentiation of immortalized human bone marrow stromal cells[48]. In our research, we found that high cholesterol downregulated the expression of LRP3. As a member of the low-density lipoprotein receptor gene family, LRP3 is a lipid endocytic protein involved in receptor mediates feedback control of cholesterol synthesis and endocytosis, a general process by which cells communicate with each other through internalization of regulatory and nutritional molecules[49]. High cholesterol could down-regulate the expression of LRP3, which may be due to the existence of a negative feedback mechanism. When there is too much extracellular cholesterol, cells downregulate the expression of endocytic proteins for self-protection. This hypothesis needs further experiments to verify. In addition to the decreased expression of LRP3 in cholesterol-induced OA models, LRP3 was also downregulated in OA patient cartilage, ACLT-induced rats OA models, senescence-induced mice OA models, and inflammatory factors-induced OA chondrocyte models. This indicated that LRP3 itself also played an important biological function in the pathogenesis of OA. Then, we used LRP3 knockdown and overexpression lentiviral vectors and *Lrp3*[−/−] mice to confirm that LRP3 can regulate the cartilage ECM metabolism, thereby regulating the process of OA cartilage degeneration. More importantly, we have confirmed that overexpression of LRP3 can effectively delay the severity of OA through intra-articular injection. Therefore, our research not only expands the biological functions of LRP3 in OA but also provides a target for the treatment of OA.

In order to explore the molecular mechanism of LRP3 in OA cartilage degeneration and screen downstream molecular targets, RNA-seq analysis was performed in this study. According to the results of the gene expression difference list and the literature search, we found that the expression of SDC4 was significantly increased after knocking down LRP3 in chondrocytes. As a polysaccharide of the heparan sulfate transmembrane transporter, the high expression of SDC4 is closely related to OA cartilage degeneration. Barre et al. first discovered that SDC4 mRNA levels were significantly increased in cartilage severe damage zone compared with normal area[50]. The expression of SDC4 was also observed in chondrocyte clusters in the superficial zone of OA knee and culture supernatant of OA chondrocyte culture[51]. Through the further reading of the literature, we found that TNF-α and IL-1β can upregulate the expression of SDC4 in cartilage tissue through the NF-κB pathway, activate ADAMTS-4, -5, and upregulate MMPs, thereby causing the degradation of ECM and leading to cartilage degeneration[52]. Frank et al. found that SDC4 was significantly increased in mouse OA cartilage and caused cartilage degeneration by activating ADAMTS-5 and upregulating MMP-3[31]. These results also have been confirmed in our research. In addition, after knocking down LRP3, the RNA-seq analysis showed that the Ras/Raf/MEK/ERK signaling pathway had the most significant changes among all activated signaling pathways. According to previous literature, the activation of the Ras/Raf/MEK/ERK signaling pathway is not only related to the occurrence and development of OA but also closely related to the regulation of SDC4[31,32]. In our study, we further found that the knockdown of LRP3 upregulated the expression of SDC4 by activating the Ras/Raf/MEK/ERK signaling pathway. More importantly, through the dual gene intervention of LRP3 and SDC4, we identified SDC4 as the downstream molecular target of LRP3 in the regulation of cartilage ECM metabolism. To date, we constructed a new interaction between LRP3 and SDC4 in OA pathogenesis.

The current study has several limitations. First, when we studied the downstream molecular mechanism of LRP3, we only searched the gene changes in chondrocytes caused by LRP3 knockdown but did not study the gene changes caused by overexpression of LRP3, which may ignore some other molecular targets. Second, we only used small

animal models to study LRP3 therapeutic effects of OA, the effect of LRP3 on large animals needs to be further investigated. We will continue to study these contents deeply in the future.

In conclusion, we found that high cholesterol inhibited the expression of LRP3 in chondrocytes, triggering an imbalance in cartilage ECM metabolism, and leading to the development of OA cartilage degeneration. Among them, SDC4 acted as a downstream molecular target of LRP3 to regulate chondrocyte ECM degeneration. And targeted overexpression of LRP3 can efficiently delay the degeneration of OA cartilage. In brief, the cholesterol−LRP3−SDC4 axis plays a vital role in OA pathogenesis and may provide a new therapeutic avenue for treating OA.

## Methods

### Clinical specimen preparation and collection
The clinical data of the patients ($n = 191$), including BMI, TC, LDL, etc., were obtained from the Department of Sports Medicine, Peking University Third Hospital. Human OA cartilage samples ($n = 21$, 14 females and 7 males with mean age of 64.7 years) were excised from patients undergoing TKA. Normal human cartilage samples ($n = 5$, 2 females and 3 males with mean age of 53.3 years) were isolated from the knee joints of donors of trauma patients. All protocols were approved by the Ethics Committee of Peking University Third Hospital (protocol number: PTTH-MEC-SOP-08-1.0-A02). Informed consent was obtained from all patients. The cartilage samples were used for subsequent histological and immunohistochemical assessments.

### HCD and surgically induced OA in C57BL/6 mice model
In this study, a surgically induced OA model used generated using the anterior cruciate ligament transection (ACLT) protocol. All animal experimental procedures were approved by the Experimental Animal Ethics Committee of Peking University Third Hospital. A total of 20 wild-type male C57BL/6 mice (8 weeks old) were provided by Beijing Vital River Laboratory Animal Technology Company and maintained in a specific pathogen-free (SPF) environment. Ten mice were fed an AIN-93G diet (Beijing Keao Xieli Feed Co., Ltd., Beijing, China) as the regular diet (RD), while the remaining were fed a modified AIN-93G diet supplemented with 2% cholesterol as an HCD. After 7 weeks on respective diets, ACLT surgery was performed on their right knees and sham surgery was performed on the left knees. Briefly, under general anesthesia, the joint capsule was cut off, and the anterior cruciate ligament of the right knee was transected. The sham operation on the contralateral knee did not involve ligament transection. The mice were sacrificed four weeks after ACLT/sham surgery for further analysis.

### Primary rat chondrocyte isolation and culture
Primary rat chondrocytes were isolated from cartilage fragments dissected from the femoral heads of 6-week-old Sprague−Dawley (SD) rat ($n = 35$). The rat cartilage fragments were mechanically sliced into $2-5\ mm^3$ pieces and enzymatically digested with 0.2% type II collagenase (Gibco, CA, USA) at 37 °C for 6 h. After digestion, the cells were resuspended in Dulbecco's modified Eagle's medium (DMEM; Gibco) with 10% (v/v) fetal bovine serum (FBS; HyClone, Logan, UT, USA) containing 1 g/L penicillin−streptomycin (Invitrogen, CA, USA) at 37 °C in a humid environment with 5% $CO_2$.

Chondrocytes were starved for 6 h without FBS and subsequently treated with IL-1β (10 ng/mL; 211-11B; PeproTech, Beijing, China) or TNF-α (20 ng/mL; 315-01B; PeproTech) for 6−48 h according to different experimental requirements.

### Cholesterol intervention of cartilage/chondrocytes in vitro
Exogenous cholesterol was delivered to the cartilages/chondrocytes using the cholesterol:methyl-b-cyclodextrin (Chol:MbCD) complex (#SLCB4694; Sigma-Aldrich, MO, USA), which makes cholesterol water-soluble and is widely used to successfully deliver cholesterol in a

variety of cell types. Human articular cartilage disks were treated with Chol:MbCD complex (50 μg/mL) and/or TNF-α (20 ng/mL) for 14 d. Similarly, primary p1 rat chondrocytes were treated with Chol:MbCD complex (50 μg/mL) and/or TNF-α (20 ng/mL) for 3 d (RNA extraction) or 5 d (protein extraction). Chondrocytes incubated with phosphate-buffered saline (PBS, vehicle) without Chol:MbCD treatment served as controls.

### RNA extraction and real-time qPCR

Total RNA was extracted from primary cultured chondrocytes using TRIzol reagent (Invitrogen, Carlsbad, CA, USA). Purified RNA (2 μg) was reverse-transcribed using a RevertAid First Strand cDNA Synthesis Kit (Thermo Fisher Scientific, Boston, MA, USA). Real-time qRT-PCR was performed using the Applied Biosystems StepOnePlus Real-Time PCR System (Foster City, CA, USA) with SYBR Green PCR Master Mix (Toyobo, Japan). Target mRNA expression levels were normalized to that of 18 S ribosomal RNA (Rn18s). The relative gene expression was calculated using the $2^{-\Delta\Delta CT}$ method and expressed as fold-change. Each RT-qPCR was performed in triplicate, with at least three different biological replicates. Primer sequences are shown in Supplementary Table 2.

### Protein extraction and western blot analysis

Chondrocytes were lysed in radioimmunoprecipitation assay (RIPA) lysis buffer containing protease inhibitors and/or a phosphatase inhibitor cocktail. The total cell lysates were prepared in lysis buffer (150 mM NaCl, 1% Nonidet P-40, 50 mM Tris, 5 mM NaF), separated by SDS polyacrylamide gel electrophoresis (PAGE), and transferred to a polyvinylidene fluoride (PVDF) membrane. After blocking with 5% non-fat dry milk in 0.1% Tween 20 TBS (TBST), the membranes were incubated with the corresponding primary antibodies overnight at 4 °C. After washing three times with TBST, the membranes were incubated with secondary antibodies at 21 °C for 1 h and visualized using the BIO-RAD ChemiDoc XRS + system.

Proteins were analyzed using antibodies against COL2A1 (ab34712; Abcam, CA, USA; 1:1000), GAPDH (TA-08; Zsjqbio, China; 1:1000), SOX9 (ab185966; Abcam, 1: 2000), LRP3 (PA5-50051; Invitrogen, USA; 1:500), SDC4 (ab74139; Abcam; 1:1000), Ras (#3965; Cell Signaling Technology, USA; 1:1000), c-Raf (#9422; Cell Signaling Technology; 1:1000), phospho-Erk1/2 (#4370; Cell Signaling Technology; 1:1000), ERK1/2 (#4695; Cell Signaling Technology; 1:1000), phospho-MEK1/2 (#9154; Cell Signaling Technology; 1:1000), MEK1/2 (#9422; Cell Signaling Technology; 1:1000), and Adams-5 (ab41037; Abcam; 1:1000). Anti-mouse (ZB-2305, HRP-conjugated) and anti-rabbit (ZB-2301, HRP-conjugated) secondary antibodies were purchased from ZSGB-BIO (Beijing, China; 1:1000).

### Histology and immunohistochemistry (IHC) assessment

Human cartilage samples were fixed in 4% neutral-buffered paraformaldehyde (PFA; Solarbio, Beijing, China), embedded in paraffin, and sectioned continuously (6-μm thick). The cartilage sections were stained with safranin O (Solarbio) for 5 min and toluidine blue (Solarbio) for 30 min, washed three times with PBS, dried, and mounted.

The excised whole knee joints of rats/mice were fixed in 4% neutral-buffered PFA for 3 d. The specimens were then decalcified for 2 d in a decalcification solution (ZLI-9307; ZSGB-BIO) and dehydrated in a graded ethanol series. Thereafter, the samples were embedded in paraffin, cut into 6-μm thick sections, and stained with Modified Safranin O-Fast Green FCF cartilage Stain Kit (#G1371; Solarbio) according to the manufacturer's instructions. Three blinded observers scored the histopathological changes in osteoarthritic cartilage using the Osteoarthritis Research Society International (OARSI) grading system[53,54].

For immunohistochemical staining (IHC), the paraffin-embedded sections were incubated with 3% $H_2O_2$ for 15 min to inhibit endogenous peroxidase, followed by incubation with 10% goat serum for 1 h at 21 °C to block non-specific antigens. Next, the sections were incubated with specific primary antibodies against COL2A1 (ab34712; Abcam; 1:100), LRP3 (human: ab115197; Abcam; 1:50) (rat/mouse: TA323023; Origene, USA; 1:50), and SDC4 (ab74139; Abcam; 1:50), and ARGSVIL (ab3773; Abcam; 1:50) overnight at 4 °C. Subsequently, the sections were incubated for 1 h at 21 °C with horseradish peroxidase-conjugated secondary antibodies (PV-6001 and PV-6002, ZSGB-BIO). The integrated optical density (IOD) value of positive staining was evaluated using ImageJ software (National Institutes of Health, MD, USA). The positive and negative controls for all antibodies are provided in figure S9.

### Immunofluorescence analysis

TNF-α- and IL-1β-treated rat chondrocytes were rinsed in PBS and then fixed with 10% neutral-buffered formalin for 30 min at 21 °C. Cells were incubated with Triton X-100 (Beyotime Biotechnology, Beijing, China) for 10 min to penetrate the cell membrane and donkey serum (Beyotime Biotechnology) for 1 h to block nonspecific binding sites. Cultured cells were incubated with primary antibodies against type II collagen (ab34712; Abcam; 1:100), SOX9 (ab185966; Abcam; 1:100), and LRP3 (sc-373736; Santa Cruz Biotechnology, USA; 1:50) at 4 °C overnight. The cells were then washed with PBS three times and incubated with Alexa Fluor 488-conjugated goat anti-rabbit (A-11008, Thermo Fisher Scientific; 1:200) or Alexa Fluor 594-conjugated goat anti-mouse (A-11005, Thermo Fisher Scientific; 1:200) secondary antibodies for 1 h at 21 °C. Nuclei were stained with DAPI (Beyotime Institute of Biotechnology, Jiangsu, China) for 10 min. Finally, the samples were rinsed with PBS and visualized using a confocal microscope (Olympus Life Science, Tokyo, Japan).

### Lentivirus construction and transfection

Lentiviral particles expressing rat/mouse LRP3 (Lv-Lrp3), rat SDC4 (Lv-Sdc4), rat LRP3 knockdown shRNA (Lv-siLrp3), rat SDC4 knockdown shRNA (Lv-siSdc4), and negative control lentiviruses (Lv-con335 and Lv-con313) were purchased from GeneChem Co. Ltd. (Shanghai, China). To construct Lv-Lrp3 and Lv-Sdc4, the mouse LRP3 (NM_001024707), rat LRP3 (NM_053541), and SDC4 (NM_012649) coding regions were cloned into GV492 vectors. The LRP3-GV492 and SDC4-GV492 vectors were mixed with pHelper 1.0 and pHelper 2.0 plasmids, and then co-transfected into HEK-293T cells with Lipofectamine™ 2000 (Invitrogen, Shanghai, China) according to the manufacturer's instructions. After 48-h transfection, viral supernatants were collected, filtered through 0.45-μm polyvinylidene fluoride membranes, and then centrifuged. The lentivirus vector system for packaging Lv-siLrp3 and Lv-siSdc4 was composed of pGCSIL-GFP vector, stably expressing shRNA and a marker (GFP fusion protein), pHelper 1.0 (gag/pol element), and pHelper 2.0 (VSVG element). The shRNA targeting rat LRP3 (5′-CACCAACTGCAGCTGGTACAT-3′; 5′-AGGCAGTTTCTACGGTTCCTT-3′; 5′-ATGCGGACTGCTGCTTGTTAT-3′), rat SDC4 (5′-ACCCTTGGTGCCACTAGATAA-3′; 5′-AGGCAGTTAC-GACTTGGGCAA-3′; 5′-TCGGGTTCCGGAGATCTAGAT-3′), and control shRNA used as a negative control (5′-TTCTCCGAACGTGTCACGT-3′) were designed, synthesized, and cloned into the pGCSIL-GFP vector by GeneChem Co. Ltd. The pGCSIL-shRNA-GFP, pHelper 1.0, and pHelper 2.0 were mixed and transfected into 293 T cells using Lipofectamine™ 2000 according to the manufacturer's instructions. After 48-h transfection, viral supernatants were collected, filtered through 0.45-μm polyvinylidene fluoride membranes, and centrifuged.

Chondrocytes (50% confluent) were transduced with lentiviral particles at a multiplicity of infection (MOI) of 100. Culture media were replaced with normal media without lentivirus 24 h after transduction. After 3 d, the green fluorescence in the cells was observed using a fluorescence microscope (Olympus Life Science, Tokyo, Japan). At >80% positive fluorescence rate and >90% cell viability, cellular mRNA and proteins were extracted for further analysis.

### Pellet culture of rat chondrocytes

Normal rat chondrocytes were transfected with Lv-con313 and Lv-siLrp3 for 3 d. The TNF-α-induced OA rat chondrocytes were transfected with Lv-con335 and Lv-Lrp3 for 3 d. Then, $1 \times 10^6$ chondrocytes from each group were added to 15-mL polypropylene conical tubes and centrifuged at 400 g for 5 min to form a cell pellet. The cells were re-suspended in chondrogenic low-glucose DMEM supplemented with 10 ng/mL recombinant human transforming growth factor-beta 3 (Cyagen Biosciences Inc., Santa Clara, CA, USA). The culture media were changed every two days. After 2 weeks of culture, the pellets were harvested for histological assessment.

### ELISA assay

ARGSVIL production in the supernatants was quantified by ELISA kits according to the manufacturer's protocol. The ARGSVIL ELISA kits were from R&D Systems (Minneapolis, MN).

### Alcian blue staining and glycosaminoglycans (GAG) content analysis

After lentiviral transfection, rat chondrocytes were fixed in 4% neutral-buffered PFA for 15 min and then stained with Alcian blue (Cyagen Biosciences Inc.) for 30 min according to the manufacturer's protocol. The total GAGs extracted from the chondrocytes were measured using the 1,9-dimethylmethylene blue (DMMB; Sigma-Aldrich) colorimetric assay. The chondrocyte samples in 96-well plates ($n = 4$ per group) were digested with 20 μL papain lysates (125 mg/mL papain, 5 mM L-cysteine, 100 mM Na2HPO4, 5 mM EDTA, pH 6.2) (Sigma, St. Louis, MO, USA) at 65 °C for 1 h and diluted in DMMB working solution (200 μL). Absorbance was then measured at 525 nm using a microplate spectrophotometer (Multiskan Sky, Thermo Fisher Scientific). The proteoglycan content in each sample was calculated based on a standard curve generated using serial dilutions of chondroitin sulfate (Sigma-Aldrich). As for DNA content assay, the chondrocyte samples in 96-well plates ($n = 4$ per group) were also digested with 20 μL papain lysates at 65 °C for 1 h, then the 20 μL specimen digested above was reacted with 200 μL of Hoechst 33258 working solution (2 μg/mL) shielded from light at 37 °C for 1 h. The intensity was measured by fluorimetry on a plate reader with an excitation of 360 nm and an emission of 460 nm. The readings were compared with the standard curves of calf thymus DNA (Sigma). Finally, when the experimental group and the control group were compared, the form of GAG/DAN was used to eliminate the difference caused by the different number of sample cells.

### C57BL/6-Lrp3 global knockout mice (Lrp3$^{-/-}$ mice)

A total of 90 C57BL/6-Lrp3 global knockout mice (Lrp3$^{-/-}$ mice) were purchased from Cyagen Biosciences, Inc. (Jiangsu, China) and were created by CRISPR/Cas9-mediated genome engineering. Eight exons of Lrp3 (NM_001024707; Ensembl: ENSMUSG00000001802) were identified, with the ATG start codon in exon 1 and the TGA stop codon in exon 8 (Transcript: ENSMUST00000122409). Exons 4–7 were selected as the target sites. Then, Cas9 and gRNA were co-injected into fertilized eggs to generate Lrp3$^{-/-}$ mice. The pups were genotyped using PCR, followed by sequencing analysis. The housing conditions for Lrp3$^{-/-}$ mice are maintained in an SPF environment. The ambient temperature is maintained at 18–22 °C, the relative humidity is maintained at 50% − 60%, and the average lighting time is 10–14 h.

OA was induced by ACLT in the 8-week-old male wild-type C57BL/6 or Lrp3$^{-/-}$ mice as described previously. The ACL of the right knee was transected and confirmed by a positive anterior drawer test result. Knee samples were harvested at 4 and 8 weeks after ACLT or sham operation. To study naturally occurring OA, Lrp3$^{-/-}$ mice were raised for 12 months (senescence-induced OA model). To evaluate the gene rescue effect on the ACLT-induced Lrp3$^{-/-}$ mice OA model, either 20 μL Lv-Lrp3 or Lv-con335 ($1 \times 10^6$ lentiviral particles) was injected into the knee joints once a week for four weeks. The wild-type and knockout mice were sacrificed for further analyses. All animal experiments were approved by the Animal Care and Use Committee of Peking University Health Science Center and were conducted in compliance with their guidelines.

### Intra-articular delivery of lentivirus in SD rat OA model

Intra-articular injections of lentiviral particles were administered 14 d after ACLT in 8-week-old male SD rats. To study the effect of local LRP3 knockdown in the knee joint cavity of ACLT-induced OA rat model, 50 μL Lv-siLrp3 or Lv-con313 ($1 \times 10^6$ lentiviral particles) was injected into the knee joints once a week for four weeks. To evaluate the therapeutic effect of upregulated LRP3 expression in the cartilage of these rats, 50 μL Lv-Lrp3 (low dose: $1 \times 10^5$ lentiviral particles; high dose: $1 \times 10^6$ lentiviral particles) or Lv-con335 ($1 \times 10^6$ lentiviral particles) was injected into the knee joints once a week for six weeks. Similarly, to evaluate the effect of local upregulation of cartilage SDC4 expression, 50 μL Lv-Sdc4 or Lv-con335 ($1 \times 10^6$ lentiviral particles) was injected into the knee joints once a week for four weeks. Rats were sacrificed one week after the last injection. The protocol was approved by the Animal Care and Use Committee of the Peking University Third Hospital. The experiments were conducted in accordance with appropriate international guidelines and all relevant ethical regulations for animal testing and research.

### Anesthesia of experimental animals

All experimental animals in this work were anesthetized by intraperitoneal injection of drugs before surgery or operation to reduce the pain. The drug for intraperitoneal injection is the compound solution of pentobarbital sodium and chloral hydrate. The specific formula is as follows: sodium pentobarbital: 0.886 g, chloral hydrate: 4.25 g, magnesium sulfate: 2.12 g, absolute alcohol: 14.25 ml, propylene glycol: 33.8 ml, and deionized water to a constant volume of 100 ml. The specific dosage is 0.25 ml/100 g for rats or mice.

### Monitoring of experimental animals

All experimental animal monitoring in this study was conducted by the Department of Laboratory Animal Science of Peking University Health Science Center.

### Hot plate test

The pain response in the joints of rats/mice in different experimental groups were evaluated by the hot plate test. Briefly, the animals were placed on a hot plate meter (Ugo Basile SRL, Italy) at 55 °C, and the time from the point when both hind limbs touched the hot plate until the appearance of hind limb responses, such as shaking, jumping, or licking, was measured. Each animal was tested three times with 15 min intervals, and the average of the three response times was considered the final pain threshold of the rat/mouse. The observers were blinded to the experimental treatments.

### Weight-bearing test

The weight distribution of the hind paws of rats was measured using an incapacitance tester (Ugo Basile SRL). The rats were placed in the detection chamber, and the left and right hind paws were placed on the pressure sensor for 9 s, after which the pressure transducer data were read. The results are shown as the difference between the weight placed on the contralateral sham (left) and the ACLT (right) hind limb. The tests were performed three times for each rat. The observers were blinded to the experimental treatments.

### Nanoindentation analysis

Biomechanical characteristics of rat knee cartilage tissue were analyzed using an in situ nanomechanical test system (TI-900 TriboIndenter, Hysitron, Minneapolis, MN, USA). Cartilage samples were

collected from femoral condyles of the sham, vehicle, and Lv-Lrp3-treated (low and high dose) ACLT-induced OA rat groups ($n = 5$/group) at 6 weeks. PBS solution was used to maintain cartilage hydration. The indentation cycle consisted of a 10 s peak load, 2 s hold, and 10 s unload. The maximum indentation depth is 2000 nm. The hardness and elastic modulus were determined from the load-depth curve.

## Micro-computed tomography (CT) analysis for knee joints
Osteophyte development and subchondral bone remodeling in rats/mouse knee joints were evaluated by micro-CT. The ACLT-induced OA model rats were sacrificed at the designated time points, and intact knee joints were excised from the surrounding soft tissue (skin and muscles). Samples ($n = 4$ per group) were scanned using micro-CT (Siemens Inveon MM Gantry, Berlin, Germany) at an isotropic resolution of 5 μm³. A three-dimensional (3D) model was constructed using Mimics Research software. Histomorphometric analysis was performed on longitudinal images of the femur subchondral bone, which was also analyzed using the Inveon Research Workplace software (Siemens Inveon MM Gantry, Berlin, Germany), determining the trabecular ratio of bone volume to tissue volume (BV/TV) and trabecular bone pattern factor (Tb. Pf.). Briefly, we created 3D analyses of the trabecular bone. The region of interest (ROI) of the trabecular bone was drawn beginning from 5% of the femur length proximal to the distal metaphyseal growth plate and extending proximally for another 5% of the total femur length. All the ROI was segmented manually on a slice-by-slice basis. The BV/TV and Tb. Pf. were collected from the 3D analyses data and used to represent the trabecular bone parameters.

## RNA sequencing (RNA-seq) for the chondrocyte/cartilage transcriptome
We performed mouse knee cartilage tissue and rat chondrocyte RNA-seq analysis using the NovelBrain Cloud Analysis Platform. Cartilage tissues were collected from the medial and lateral femoral condyles of wild-type C57BL/6 mice on RD or HCD after seven weeks. Rat chondrocytes and the control group were transfected with Lv-siLrp3 and Lv-con313 (empty vector), respectively. After 3-d transfection, total RNA was extracted from the rat chondrocytes and cartilage tissue using TRIzol reagent. cDNA libraries were constructed for each pooled RNA sample (rat and mice) using VAHTSTM Total RNA-seq (H/M/R). Differential gene and transcript expression analyses were performed using TopHat and Cufflinks. HTseq was used to count the gene and lncRNA counts. The FPKM method was used to determine gene expression. We applied the DESeq algorithm to identify the differentially expressed genes. Significant analysis was performed using the $P$-value and false discovery rate (FDR) analysis. Differentially expressed genes were considered as those with a fold change >2 or fold change <0.5, FDR < 0.05. GO analysis was performed to elucidate the biological implications of the differentially expressed genes, including biological process (BP), cellular component (CC), and molecular function (MF). GO annotations from NCBI (http://www.ncbi.nlm.nih.gov/), UniProt (http://www.uniprot.org/), and Gene Ontology (http://www.geneontology.org/) were downloaded. Pathway analysis was used to identify the significantly influenced pathways in which the differentially expressed genes were affected according to the KEGG database. Fisher's exact test was used to identify the significantly influenced GO categories and pathways. The threshold of significance was defined by the $P$ value.

## Filipin staining for chondrocyte cholesterol assay
Cholesterol in chondrocytes was stained using the Cell Cholesterol Filipin Staining Kit (GMS80059.1; GenMed Scientifics, USA) according to the manufacturer's instructions. Briefly, rat chondrocytes were washed with PBS, fixed in 10% neutral-buffered formalin for 10 min at 21 °C, and then incubated with Filipin Staining solution for 30 min at 21 °C in the dark. After washing three times with PBS, the chondrocytes were observed using a confocal microscope (Olympus Life Science).

## Co-immunoprecipitation
For co-immunoprecipitation assays, chondrocytes transfected with Lv-Sdc4 (MOI = 100) to overexpress SDC4 were lysed with a cold RIPA lysis buffer. Lysates were centrifuged at 12,000$g$ for 10 min, and the supernatants were collected. Supernatants containing proteins were then incubated with 1 μg anti-SDC4 (PA1-32485; Invitrogen, USA), or 1 μg normal rabbit IgG (ab172730; Abcam) antibodies overnight at 4 °C. Protein lysates were subsequently incubated with 20 μL pre-washed protein A/G PLUS-Agarose beads (sc-2003; Santa Cruz Biotechnology, USA) with gentle rotation for 3 h at 4 °C. The immunoprecipitates were then analyzed by western blotting as described above using primary antibodies against ADAMT-5 (ab41037; Abcam) to detect SDC4 protein.

## Statistical analysis
All statistical analyses were performed using SPSS (version 20.0; IBM Corp, Chicago, IL, USA). Data are presented as mean ± standard deviation. The Shapiro–Wilk test was used for testing data normality, and Levene's test was used for testing homogeneity of variance. The Student's $t$-test or Mann–Whitney rank sum test (for non-parametric data) was performed to compare two independent groups. For comparison between multiple values, one-way analysis of variance (ANOVA) or Kruskal–Wallis test (for non-parametric data) with Tukey's post hoc test to assess differences between specific groups. $n$ indicates the number of biologically independent samples, mice per group, or human specimens. Significant data are indicated by *$P < 0.05$; **$P < 0.01$; and ***$P < 0.001$.

## Reporting summary
Further information on research design is available in the Nature Portfolio Reporting Summary linked to this article.

## Data availability
All data generated or analyzed during this study are included in this published article (and its supplementary information files). The source data for Figs. 1–7 and Supplementary figures generated in this study are provided in the Source Data file. The raw RNA sequencing data generated in this study have been deposited in the National Center for Biotechnology Sequences Read Archive under accession code ID PRJNA855888. Source data are provided in this paper.

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

## Acknowledgements

This work was supported by the National Natural Science Foundation of China (No. 82072486, X.Q.H.; No. 81972101, Y.F.A.; No. 82002346, Y.Y.S.; No.32000923, J.C.) and Beijing Municipal Natural Science Foundation (No. 7222213, X.Q.H.; No. 7214304, J.C.). We would like to acknowledge Dr. Tong Wu, Dr. Junyan Wang, La Li Ph.D. and Muyang Sun Ph.D. at Peking University Third Hospital, and Dr. Shan Wang at Beijing Friendship Hospital for experimental technical guidance. We would also like to thank Editage (www.editage.com) for English language editing.

## Author contributions

C.X.C. conducted the majority of the experiments and completed the paper. Y.Y.S. participated in the experiment and helped with paper preparation. YFA, X.Q.H. and J.C. conceived the project and designed the experiments. X.Z., Q.L., and J.H.Z. helped perform the animal surgery. F.Y.Z. and Q.Y.M. collected the clinical specimens. W.L.D. and Z.L.L. collected animal samples and scoring. X.N.D. analyzed and interpreted the data. J.Y.Z., X.F., J.Y.Z., and W.Q.Y. revised the paper. Y.F.A. and X.Q.H. supervised the project. C.X.C. and Y.Y.S. contributed equally to this work.

## Competing interests

The authors declare no competing interests.
