## [Peer Review File · Nature Communications]

Cholesterol-induced LRP3 downregulation promotes cartilage degeneration in osteoarthritis by targeting SDC4REVIEWER COMMENTS

Reviewer #1 (Remarks to the Author):

Manuscript#: NCOMMS-21-09195

Manuscript Title: Cholesterol-induced LRP3 downregulation promotes cartilage degeneration in osteoarthritis by targeting SDC4

Overview: In this study the authors present original data to suggest that the cholesterol-LRP3-SDC4 axis plays critical roles in the OA development, and the LRP3 gene therapy may provide a new therapeutic option for OA treatment.

Specific comments: This paper includes new and important information. However, some of the writing is very poor and requires heavy and careful editing. It is important to state that the quality of the writing is actually very poor and this really needs to be addressed. Numerous inappropriate and incorrect statements have been made in this manuscript. This is totally unacceptable. This reviewer has provided a short list of some of the most important revisions that will be required.

Title: the title is clear and concise.

Abstract: the first sentence of the abstract is misleading. Emerging evidence suggests that osteoarthritis (OA) is associated with high cholesterol levels in some but not all OA patients. Stating that high cholesterol levels are present in all OA patients is incorrect and unfortunately overstated by the authors. It is highly likely that there is a metabolic OA phenotype that is associated with high cholesterol levels. It does not mean that all OA patients have high cholesterol levels. There is no mention of the metabolic phenotype in the abstract and the rest of the paper and this is an important weakness that must be addressed by the authors. The anterior cruciate ligament transection (ACLT)-induced OA progression in rats is simply a model of OA and does not represent the totality of the human OA condition.

Introduction: Lines 52-52 contain grammatical and scientific errors and should read: New evidence suggests that metabolic syndrome is associated with an increased risk of OA, especially lipid metabolism, which points to the existence of a metabolic OA phenotype 5.

Lines 57-58 should read: Meanwhile, several lines of evidence have demonstrated that some OA patients have significantly higher serum levels of low-density lipoprotein (LDL) compared to healthy controls 8, 9. It is important to make it clear that some OA patients exhibit these characteristics. This is certainly not true of all OA patients. The authors should go through the introduction carefully and make sure that they are not overstating facts just because it strengthens their hypothesis. Every sentence needs to be checked for contextual accuracy.

Discussion: the first paragraph of the discussion contains several sentences that need to be edited carefully for correct use of scientific English.

In lines 300-301 the authors state: The pain and OA severity scores of OA patients with hypercholesterolemia were also higher than those of the control group 29. This is a clear indication that this phenomenon occurs mainly in patients with co-morbidities such as hypercholesterolemia. This does not mean that all patients with OA have hypercholesterolemia and metabolic disease.

Metabolic OA is a phenotype and only a proportion of all OA patients have this form of OA.

Lines 310-311: These results all support the notion that high cholesterol make a significant contribution to the development, progression, and severity of OA. Unfortunately this is incorrect and overstated.

The authors should humbly accept that their data has enhanced our understanding of some of the key players involved in a metabolic phenotype of OA that implicates hypercholesterolemia and metabolic disease. Stating that all OA patients have hypercholesterolemia and metabolic disease is incorrect and highly misleading. The data presented is original and exciting but the language used to describe the data is overall inaccurate, overstated and unacceptable.

Other comments:

1. The number of human patients is very low for such a study.
2. Lines 413: Cholesterol intervention of cartilage/chondrocytes in vitro - this section is highly experimental and does not prove that cholesterol addition in vitro can recapitulate the in vivo situation in hypercholesterolemia.
3. There are spelling errors in some of the figures. For example, human is incorrectly spelled in figure 1 panel I.
4. All the western blots should be submitted without cropping as separate full figures in the supplementary data. These blots must be uncropped and properly labelled to indicate molecular weight markers.
5. The LRP3 westerns look highly suspect in figure 3 panels b and g. The full western blots must be provided as supplementary data.

Reviewer #2 (Remarks to the Author):

In this study, the authors found that i) expression of LRP3 gene and protein levels are decreased during the OA development, ii) deficiency of LRP3 aggravates the degeneration of knee joint cartilage, iii) overexpression of LRP3 in cartilage attenuates OA progression, and iv) LRP3 knockdown upregulates SDC4 expression by activating the Ras signaling pathway.

This study was well designed and the sufficient evidence was provided to claim existence of cholesterol-LRP3-SDC4 in the knee cartilage. As far as I know, this is the first study demonstrated a role of LRP3 in cartilage pathophysiology. However, there are some methodological caveats to this study.

1) In Fig 6, aggrecanase activity was investigated by Western blot analysis and immunohistochemistry (IHC) using aggrecan neopeptide anti-ARGSV antibody. Deglycosylation of chondroitin sulfate and keratan sulfate are required for the WB analysis due to the size of aggrecan proteoglycan. However, no information provided at all in the Method. Even if deglycosylation of aggrecan was properly done, as published elsewhere, ARGSV aggrecan fragments are supposed to be much larger than 70 kDa shown in this study. The lack of an entire membrane with molecular ruler did not help as well.

Aggrecan presents in cartilage via binding of its G1 domain to linker protein, which is anchored to hyaluronan. Thus, ARGSV aggrecan fragments are released into synovial fluid upon cleavage by aggrecanases. This is very well known and thereby detection of the other side of the fragments by using NITEGE neopeptide antibody in cartilage are widely used.

For these reasons, the direct effect of cholesterol-LRP3-Sdc4 axis on aggrecanase activity is not clear.

2) An entire blot membrane with a molecular ruler corresponding to each cropped western blot should be provided as a supplementary. For example, SDC4 is known to form a dimer and dimerization is suggested to be important for its function. Another example is that C-terminal ancillary domains are crucial for aggrecanolytic activity of aggrecanases including ADAMTS-5 and -4. Furthermore, these domains are important for heparin binding hence potential interaction sites for SDC4.

3) Frank et al (NatMed 2009) demonstrated that SDC4 increases activity of ADAMTS-5 not only by binding to the enzyme but also by upregulating MMP-3, which potentially activate ADAMTS-5. It is thus important to test role of LRP3 on MMP-3 expression level.

4) Cholesterol downregulates LRP3, which is one of the major findings in this study. It is thus important to discuss possible molecular mechanism behind this.

5) It is likely that the authors used PBS but not isotype control antibody as a negative control of IHC

analysis of LRP3, SDC4 and COL2. LRP3 IHC was validated using LRP3 KO tissue but SDC4 and COL2 staining need to be validated using isotype control or tissue sections of SDC4 or COL2 null mice.

6) Supplementary Fig 2 is important as this first demonstrated a link between high cholesterol and LRP3 levels. I would suggest to show in main figure. On the other hand, Fig 2 e-n can be supplementary as these experiments are similar to the experiments as Fig 1 m-f.

7) Fig 3 d and i, not only cell morphology but also cell population looks quite different. The difference of alcian blue staining is mainly due to less cells? Dose LRP3 affect cell growth and viability?

8) Possible function of LRP3 should be discussed. It is not clear at all about how is LRP3 involved in Ras signaling pathway or SDC4 expression.

Reviewer #3 (Remarks to the Author):

Major conceptual issue is the seriously misleading statement (L54) that "hypercholesterolemia has been proved to be an independent risk factor for systemic OA." A reference from 2003 supporting this claim is cited: "Al-Arfaj AS. Radiographic osteoarthritis and serum cholesterol. Saudi Med J 24, 745-747 695 (2003)." In contrast, the recent (Sep 22, 2020) data of Schwager et al (doi: 10.1002/acr.24455.) is neither cited here nor discussed. It studied participants from the Multicenter Osteoarthritis study (MOST) cohort at risk of developing knee OA from baseline through 7 years. The data revealed that neither total cholesterol, LDL nor HDL showed a significant association with radiographic or symptomatic OA. Additionally, they found no association of these lipid measures with cartilage loss, worsening synovitis or worsening knee pain, rejecting a possible association between total cholesterol, LDL or HDL with OA outcomes. As this is the clinical basis of the work, this ignorance of serious published work is deeply troubling.

Logical issue #1: Figure 1 E, OARSI grade suggests that a high cholesterol diet already induces OA without any additional ACLT, shown by the significant differences in MFC and MTP. From a clinical point of view this would mean that patients with a high cholesterol diet would have an increased incidence of OA, which is not supported by clinical studies. Moreover, the data also suggest that following ACLT, structural or a progression is worse in high cholesterol diet animals, again, a finding that is not supported by clinical data.

Logical issue #2: If the 20 patients (Supplementary Figure 1) with OA have a significantly higher BMI, then the BMI-induced overload could play a role in OA, not the associated hyperlipidemia.

Major technical issue #1: P4 L91 "which indicated that the hyperlipidemia was significantly positively correlated with the development of OA (Fig. 1a)." No correlation analysis was shown on Fig1a or in the whole study, so there is no evidence for this statement. This reviewer would expect from authors submitting such a sophisticated analysis to know the difference between an "association" and true "correlation".

Major technical issue #2: Proteoglycans and type II collagen are measured by IOD value. True biochemical quantification is however standard and needed (e.g. DMMB).

Major technical issue #3: The Western blots shown in the Supplementary materials in larger size are inadequate to assess the reliability of the performed experiments.

Major technical issue #4: The statistical tests are inadequate. For comparing non-parametric data, e.g. OARSI grades (Fig1f, Fig2f,j, Fig3l,p, Fig4c), non-parametric tests (e.g. Mann-Whitney rank sum test, Kruskal-Wallis ANOVA) are needed. In the other data were there any cases when the normality

test failed? If yes, please describe which tests were used for non-normally distributed data.

Minor comments

Overall writing and logic needs to be improved in many cases as detailed below.

The Western blots shown in the Supplementary materials in larger size are still inadequate to assess the reliability of the performed experiments. In Supplementary materials please show the whole membranes of the Western blots including the whole protein ladder on each. Please also show 3 biological or technical repeats of each experiment. Finally and most importantly, please show the amidoblack staining (<https://www.ncbi.nlm.nih.gov/pmc/articles/PMC5646381/>) of the entire membranes for each experiment to be able to compare the whole protein content of the loaded samples.

P2 L51: "Previous studies largely believed that OA cartilage degeneration is related to aging², trauma³, and abnormal biomechanics⁴. The existing more evidence shows that metabolic syndrome is associated with an increased risk of OA, especially lipid metabolism⁵" This sentence is phrased so that the reader might think that these previous studies are not true or outdated, and OA is only linked to metabolic syndrome. Please rephrase the sentence and emphasize that OA is a multifactorial disease with many possible causes. Moreover, as described in references 26-27, neither metabolic syndrome nor its components, with the exception of hypertension, are associated with incident OA after adjustment for BMI or body weight.

P2 L51 Please also mention that obesity (or high BMI), which is a well-known risk factor of OA by increasing joint load, is strongly connected to metabolic syndrome.

P2 L52 "The existing more evidence". Please rephrase.

P3 L74 "Aggrecanases belong to the ADAMTS family of metalloproteinases, and the degradation of aggrecan by ADAMTS-4 and ADAMTS-5 20." Something is missing from this sentence.

P3 L78 "Here, we investigated the functional role of the cholesterol-LRP3-SDC4 axis in OA development in this study." "Here" or "in this study" is redundant.

P3 L80 "The elucidation of such mechanisms could facilitate the development of new and effective therapeutic targets for the treatment of OA." Instead of "therapeutic targets" therapies would fit better.

The BMI data are repeated in Fig1a and Supp Fig1C and the table of Supp Fig1a-b. Please present them only once.

P4 L91 "which indicated that the hyperlipidemia was significantly positively correlated with the development of OA (Fig. 1a)." No correlation analysis was shown on Fig1a or in the whole study, so there is no evidence for this statement.

P4 L93 "Safranin-O and toluidine blue staining showed a significant proteoglycans reduction in response to cholesterol and TNF- α ". Why 50 $\mu\text{g}/\text{mL}$ cholesterol concentration was applied in vitro, if the normal level in the blood is $<200 \text{ mg}/\text{dl}$ ($<2000 \mu\text{g}/\text{ml}$), and pathological high total cholesterol level starts from $>240 \text{ mg}/\text{dl}$ ($>2400 \mu\text{g}/\text{ml}$)? Is cholesterol present in the synovial fluid in obese people? If yes, in what concentration?

P4 L105 "HCD prominently down-regulated the chondrocytes anabolic genes (Col2a1, Acan and Sox9), and upregulated the anabolism gene (Mmp13 and Adamts5) (Fig. 1g)." Sox9 is not shown on the figure. Mmp13 and Adamts5 are catabolism genes.

P4 L118 "from 5 trauma patient (as control)". Patients

P4 L120 "the surface of OA cartilage displayed severely degenerated and exhibited cracks and fissures," something is missing from the sentence

P5 L126 "Senile Mice were mainly manifested by the". Strange phrasing

In Fig2m and Fig3a,f the expression of Col2, Acan, Sox9 and LRP3 is shown separately in rat chondrocytes, and TNF- α induced rat OA chondrocytes with and without LRP3 knockdown or overexpression. A combination image of the absolute values of Fig2m and Fig3a,f could also be interesting (maybe to be shown as a supplementary figure) to see the differences between the control groups and the treatment groups, and to see whether the overexpression of LRP3 in OA chondrocytes restore the levels of Col2, Acan, Sox9 and LRP3 to the normal levels.

In Supp Fig S5 it should be mentioned that vehicle is Lv-con313

P6 L162-164: Supp Fig S7 should be cited between S6 and S8. In Supp Fig S7 the body size of only the newborn mice is shown. Is there any difference in adult mice between age-matched WT and LRP3-/- ? If LRP3 mediates lipid metabolism, it might affect the adult body weight too. Also, do LRP3-KO mice gain weight in response to high-cholesterol diet similarly to WT mice shown in Fig1e?

Supp Fig S8b: the explanation for the red-blue colours is missing

P7 and everywhere else in the manuscript: All figures, figure panels and tables should be presented in order. For example, the description of panel A of figure 3 should not come before that of panel B of figure 2. Now it is Fig4a, b, f, b, d, c, e, f; also on page 8 for Fig 5.

P7 L206 "OARSI scores was" - scores were or score was

P8 L217 "up-regulated in responded to Lrp3 knockdown" - in response to

What was the specific reason to choose SDC4 for further analysis over the many other possible candidates which also showed altered expression?

P8 L228 and everywhere else: KEGG, MEK, ERK, etc.: please define the abbreviations where they are mentioned first.

P9 L249, 258 typo "western bolt"

P10 L281 "In this paper, we have made several key observations presented here." Redundant "In this paper" or "presented here."

P10 L289 "... chondrocytes. And the overexpression" Generally do not start a sentence with "and".

P10 L296 "In recent years, a large number of epidemiological and basic researches have shown that metabolic syndrome, which is dominated by abnormal cholesterol metabolism, is closely associated with the occurrence and development of OA26, 27." In fact, both references state the opposite, that neither metabolic syndrome nor its components, with the exception of hypertension, are associated with incident OA after adjustment for BMI or body weight.

P11 L303 please define "ApoE-/- mice"

P11 L312 "However, the molecular mechanism of cholesterol-induced OA cartilage degeneration has

not yet been completely eliminated." - Elucidated

P12 L359 "we identified that SDC4 as the downstream molecular target". "That" is not needed in the sentence

In Fig2, human control and OA histological sections were shown for LRP3 expression. Can the authors show the expression of SDC4 also in these samples?

P13 L377-381 The patient numbers are not matching, please clarify them: were the n=40 patients which were evaluated for clinical data also used for cartilage samples?
"OA n=20, 14 females and 7 males" their sum is 21, not 20.

P13 L388 "anterior cruciate ligament transplantation (ACLT)" - transection

P14 L403: n=? of the rats is missing

P16 L449, P17 L470 typo: "VAGSVIL" instead of ARGSVIL

P17 L464 please add reference for OARSI grading

P22 L633 "21□, and then incubated with Filipin Staining solution for 30 min at 21□ in the dark." Some characters are not displayed correctly.

Fig1c typo: collagen

Fig1l typo: humam

Fig 1l: What comparisons do the significance signs show? Are they compared to the red column (control)?

Fig2b: the two diagrams are the same. In the raw data excel file the original datasets are different, so it is probably only a mistake of inadvertence.

Fig2e: The staining of the bone and the marrow looks also faint on the ACLT image, not only that of the cartilage. Were they made with the same slice thickness, staining protocol and imaging settings? Please provide comparable images.

Fig 2g: Why is the bone marrow less stained in ACLT? Were the images made with the same slice thickness, staining protocol and imaging settings? If not, please provide comparable images.

Fig3b,g: Why is LRP3 so bad quality and pixelated in these panels while on all other Western blots they look fine?

Fig3k: Scale values are missing in some panels while on some others they are redundant.

Fig4c,d,e: typo: Hige

Fig4e: "E" is capital

Fig4g: Why are the menisci missing in vehicle and low dose? In these two groups, especially in vehicle, the subchondral bone looks extremely injured. What is the explanation of this?

Please describe the selection of the subarticular spongiosa ROIs for micro-CT measurements in more details in the Methods (how were the boundaries chosen, was it an automated algorithm or manual

drawing, etc.?).

Fig5f: what is the difference between the green and orange columns? Under both Lv-RLP3 is written.

Fig5g: typo: KRGG

Reviewer #1:

1. Comment:

The first sentence of the abstract is misleading. Emerging evidence suggests that osteoarthritis (OA) is associated with high cholesterol levels in some but not all OA patients. Stating that high cholesterol levels are present in all OA patients is incorrect and unfortunately overstated by the authors. It is highly likely that there is a metabolic OA phenotype that is associated with high cholesterol levels. It does not mean that all OA patients have high cholesterol levels. There is no mention of the metabolic phenotype in the abstract and the rest of the paper and this is an important weakness that must be addressed by the authors.

Response:

Thank you for your professional advice. This sentence is indeed prone to misunderstanding by readers, so we take your suggestion and amend the first sentence of the abstract to "**Emerging evidence suggests that osteoarthritis (OA) is associated with high cholesterol levels in some OA patients.**" It should be pointed out that we do not deliberately emphasize that all OA is related to high cholesterol. Our research aims to explore the new pathogenesis of OA, try to reveal the role of cholesterol metabolism and its related gene in the progression of OA, find molecular targets that play an important role in this process, and provide a potential theoretical basis for molecular therapy for high cholesterol-related metabolic OA phenotype.

Regarding the ignorance of the important concept of "metabolic OA phenotype", we have added and discussed it in the abstract and subsequent text according to your suggestion.

2. Comment:

The anterior cruciate ligament transection (ACLT)-induced OA progression in rats is simply a model of OA and does not represent the totality of the human OA condition.

Response:

Thank you for your professional advice. ACLT-induced OA model is a relatively common and well-recognized model, and to overcome the shortage of this model that does not represent the totality of the human OA condition, we also used human OA cartilage tissue, rat ACLT-induced OA model, and mouse natural aging-induced OA model to detect the expression of LRP3 in OA. When using Lrp3 overexpression to treat OA, we used rat ACLT-induced OA model and Lrp3 knockout-

induced OA model. Due to the limitation of the length of the abstract, we have made a detailed clarification in the main text. At the same time, we also added a corresponding description in the abstract: “Regardless of diet, LRP3 overexpression in cartilage attenuated anterior cruciate ligament transection (ACLT)-induced OA progression in rats and Lrp3 knockout-induced OA progression in mice.”

Unfortunately, we have not conducted treatment studies in other large animal OA models, which is the limitation of this study. We are currently still carrying out preliminary exploratory work, and will conduct treatment studies in more OA models in the following studies.

3. Comment:

Introduction: Lines 52-52 contain grammatical and scientific errors and should read: new evidence suggests that metabolic syndrome is associated with an increased risk of OA, especially lipid metabolism, which points to the existence of a metabolic OA phenotype.

Response:

Thank you for your suggestion. We have made a grammatical modification in the corresponding part of the article according to your suggestion and amended it to: “Furthermore, new evidence supports a link between OA and metabolic syndrome (MetS), such as hypertension, insulin resistance and dyslipidemia, which are associated with an increased risk of OA.” And we also cited recent studies on the relationship between cholesterol and OA here. At the end of the first paragraph of the Introduction section, we also added this sentence: “Above evidences indicated it is highly likely that there is a metabolic OA phenotype which is associated with high cholesterol levels.”

4. Comment:

Introduction: Lines 57-58 should read: Meanwhile, several lines of evidence have demonstrated that some OA patients have significantly higher serum levels of low-density lipoprotein (LDL) compared to healthy controls. It is important to make it clear that some OA patients exhibit these characteristics. This is certainly not true of all OA patients. The authors should go through the introduction carefully and make sure that they are not overstating facts just because it strengthens their hypothesis. Every sentence needs to be checked for contextual accuracy.

Response:

Thanks for your suggestion, sorry for the ambiguity in our description here, we revised the sentence as you suggested: “In the field of clinical research, several lines of evidence have demonstrated that some OA patients have significantly higher serum levels of low-density lipoprotein (LDL) compared to healthy controls.” We have also gone through the introduction and checked every sentence for contextual accuracy.

5. Comment:

Discussion: the first paragraph of the discussion contains several sentences that need to be edited carefully for correct use of scientific English.

Response:

Thank you for your suggestion. We have revised the first paragraph of the discussion to: “In this paper, we have made several key discoveries: (1) high cholesterol stimulation on chondrocytes caused the disorder of cartilage ECM metabolism in vitro and in vivo, leading to cartilage degradation and subsequent OA progression. During this process, the expression of cholesterol metabolism-related gene, LRP3, was significantly reduced in chondrocytes; (2) the expression of LRP3 was downregulated in degenerated human OA cartilage and several murine models of OA. Lrp3 gene deletion aggravated cartilage degeneration in mice caused by ACLT, while overexpression of LRP3 in the joint by the lentivirus significantly suppressed cartilage ECM degeneration, thereby inhibiting the progression of OA; (3) knocking-down of LRP3 upregulated the expression of SDC4 by activating the Ras/Raf/MEK/ERK signaling pathway in chondrocytes, and overexpression of SDC4 induced cartilage ECM degeneration through binding and activating ADAMTS-5. Taken together, the above observations suggested that cholesterol-LRP3-SDC4 axis play an important role for the maintenance of cartilage homeostasis and might be a therapeutic target for the treatment of OA (Fig. 7o).”

6. Comment:

Discussion: In lines 300-301 the authors state: The pain and OA severity scores of OA patients with hypercholesterolemia were also higher than those of the control group. This is a clear indication that this phenomenon occurs mainly in patients with co-morbidities such as hypercholesterolemia. This does not mean that all patients with OA have hypercholesterolemia and metabolic disease.

Metabolic OA is a phenotype and only a proportion of all OA patients have this form of OA.

Response:

Thanks again for your suggestion. Indeed, as you have described, not all patients with OA have hyperlipidemia. The descriptions we used in the article were too absolute, which caused confusion for reading. Like you said, not all the occurrence and development of OA are related to high cholesterol. We would like to point out that high cholesterol may be one of the risk factors for OA by studying high cholesterol-related OA, and to explore the potential role of cholesterol metabolism-related genes in the pathogenesis of OA. In this regard, we have gone through the manuscript and revised the overstated description to make the content expressed in the article more accurate and objective.

7. Comment:

Discussion: Lines 310-311: These results all support the notion that high cholesterol makes a significant contribution to the development, progression, and severity of OA. Unfortunately, this is incorrect and overstated. The authors should humbly accept that their data has enhanced our understanding of some of the key players involved in a metabolic phenotype of OA that implicates hypercholesterolemia and metabolic disease. Stating that all OA patients have hypercholesterolemia and metabolic disease is incorrect and highly misleading. The data presented is original and exciting but the language used to describe the data is overall inaccurate, overstated and unacceptable.

Response:

Thank you for your suggestion. Our description here is indeed overstated. So, as you suggested, we changed this to: “Although the relationship between high cholesterol and the occurrence and development of OA is still controversial in the clinical cohort study, based on existing clinical and basic research, it is highly likely that there is a metabolic OA phenotype that is associated with high cholesterol levels. However, the mechanisms underlying the complex interactions between lipid metabolism, especially cholesterol metabolism, and OA has not yet been completely clarified.”

8. Comment:

The number of human patients is very low for such a study.

Response:

Thank you for your valuable comments. Based on your suggestions, we re-collected the data, increased the number of OA patients to 100, and at the same time increased the number of control groups to 91, and re-analyzed the data.

9. Comment:

Lines 413: Cholesterol intervention of cartilage/chondrocytes *in vitro* - this section is highly experimental and does not prove that cholesterol addition *in vitro* can recapitulate the *in vivo* situation in hypercholesterolemia.

Response:

Thanks for your professional advice. Previous study showed that the total cholesterol in the synovial fluid of some OA patients was higher than that of healthy patients (PMID: 33870758), and this section of the experiment was to simulate the stimulation of chondrocytes by cholesterol in synovial fluid. Although it cannot completely recapitulate the *in vivo* situation of hypercholesterolemia, it can provide certain scientific evidence for this study from some perspectives. In addition, in our study, there was also an *in vivo* experimental part. We carried out corresponding research by causing hypercholesterolemia in experimental animals through high-cholesterol diet (PMID: 30728500).

10. Comment:

There are spelling errors in some of the figures. For example, human is incorrectly spelled in figure 1 panel I.

Response: Sorry for the spelling errors. We have corrected them in the corresponding figure.

11. Comment:

All the western blots should be submitted without cropping as separate full figures in the supplementary data. These blots must be uncropped and properly labelled to indicate molecular weight markers.

Response:

Thank you for your professional advice. We have repeated all the western blots results of this experiment and provided a new uncropped and original full film figures with indicate molecular

weight markers in the supplementary data.

12. Comment:

The LRP3 westerns look highly suspect in figure 3 panels b and g. The full western blots must be provided as supplementary data.

Response:

Thank you for your suggestion. We purchased a new LRP3 antibody (PA5-50051; Invitrogen, USA) and repeated the western blots experiment. We also provided new uncut film figures with molecular weight markers to make the results clearer and more convincing.

Reviewer #2:

1. Comment:

In Fig 6, aggrecanase activity was investigated by Western blot analysis and immunohistochemistry (IHC) using aggrecan neoepitope anti-ARGSV antibody. Deglycosylation of chondroitin sulfate and keratan sulfate are required for the WB analysis due to the size of aggrecan proteoglycan. However, no information provided at all in the Method. Even if deglycosylation of aggrecan was properly done, as published elsewhere, ARGSV aggrecan fragments are supposed to be much larger than 70 kDa shown in this study. The lack of an entire membrane with molecular ruler did not help as well. Aggrecan presents in cartilage via binding of its G1 domain to linker protein, which is anchored to hyaluronan. Thus, ARGSV aggrecan fragments are released into synovial fluid upon cleavage by aggrecanases. This is very well known and thereby detection of the other side of the fragments by using NITEGE neoepitope antibody in cartilage are widely used. For these reasons, the direct effect of cholesterol-LRP3-Sdc4 axis on aggrecanase activity is not clear.

Response:

Thanks for your professional opinion. For this part of the experiment, our research group re-discussed and re-designed it. Initially, we intend to follow your suggestion and use the NITEGE neoepitope antibody for WB analysis. But unfortunately, by reviewing the literature, most of the NITEGE neoepitope antibody in the literature were donated by other relevant laboratories, and we could not purchase NITEGE neoepitope antibody from antibody manufacturers (such as Abcam company, CST company, etc.). Therefore, we continued to use the ARGSVIL antibody. Indeed,

ARGSV aggrecan fragments are mainly released into synovial fluid upon cleavage by aggrecanases, so we used Elisa method to detect ARGSV aggrecan fragments in chondrocytes culture supernatant of experimental group to provide evidence for the direct effect of cholesterol-LRP3-Sdc4 axis on aggrecanase activity. The corresponding experimental results were shown in revised Fig.6f, h, l, n.

2. Comment:

An entire blot membrane with a molecular ruler corresponding to each cropped western blot should be provided as a supplementary. For example, SDC4 is known to form a dimer and dimerization is suggested to be important for its function. Another example is that C-terminal ancillary domains are crucial for aggrecanolytic activity of aggrecanases including ADAMTS-5 and -4. Furthermore, these domains are important for heparin binding hence potential interaction sites for SDC4.

Response:

Thank you for your professional advice. We have repeated all the western blots results of this experiment and provided a new uncropped and original full film figures with indicate molecular weight markers in the supplementary data, making the results clearer and more credible.

3. Comment:

Frank et al (NatMed 2009) demonstrated that SDC4 increases activity of ADAMTS-5 not only by binding to the enzyme but also by upregulating MMP-3, which potentially activate ADAMTS-5. It is thus important to test role of LRP3 on MMP-3 expression level.

Response:

Following your suggestion, we have supplemented the corresponding experiments, exploring the role of LRP3 on MMP-3 expression level. At the same time, we also repeated the experimental results of Frank (NatMed. 2009) to verify the effect of SDC4 on MMP3 expression. The corresponding experimental results were shown in Fig.6e, g, k, m.

4. Comment:

Cholesterol downregulates LRP3, which is one of the major findings in this study. It is thus important to discuss possible molecular mechanism behind this.

Response:

Thank you for your suggestion. We initially explored the possible molecular mechanism by which cholesterol downregulates LRP3 in the Discussion section: “In our research, we found that high cholesterol downregulated the expression of LRP3. As a member of the LDLR (low density lipoprotein receptor) gene family, LRP3 is a lipid endocytic protein involved in receptor mediated feedback control of cholesterol synthesis and endocytosis, a general process by which cells communicate with each other through internalization of regulatory and nutritional molecules (M S Brown et al, Science 1986, PMID: 3513311). High cholesterol could down-regulate the expression of LRP3, which may be due to the existence of a negative feedback mechanism. When there is too much extracellular cholesterol, cells down-regulate the expression of endocytic proteins for self-protection. This hypothesis needs further experiments to verify.”

5. Comment:

It is likely that the authors used PBS but not isotype control antibody as a negative control of IHC analysis of LRP3, SDC4 and COL2. LRP3 IHC was validated using LRP3 KO tissue but SDC4 and COL2 staining need to be validated using isotype control or tissue sections of SDC4 or COL2 null mice.

Response:

Thank you for your professional advice on our negative control for immunohistochemical staining. In this article, we did use PBS instead of the primary antibody as a blank control to eliminate the result error caused by the non-specificity of the secondary antibody. Subsequently, we listened to your suggestion and added an isotype control using the corresponding primary antibody's IgG instead of the primary antibody to detect the background staining caused by the non-specific binding of the antibody to the tissue cells. The corresponding results are shown in our supplementary figure 9.

6. Comment:

Supplementary Fig 2 is important as this first demonstrated a link between high cholesterol and LRP3 levels. I would suggest to show in main figure. On the other hand, Fig 2 e-n can be supplementary as these experiments are similar to the experiments as Fig 1 m-f.

Response:

Thanks for your professional advice. We combined the data in the original Supplementary figure 2 with part of the data in the original figure 1 to jointly construct a revised figure 2, which presents all the data establishing the link between high cholesterol and LRP3 expression. We also modified the original figure 2 to the revised figure 3. The data presented in this figure is mainly to establish the relationship between LRP3 and OA.

7. Comment:

Fig 3 d and i, not only cell morphology but also cell population looks quite different. The difference of alcian blue staining is mainly due to less cells? Dose LRP3 affect cell growth and viability?

Response:

Indeed, as you pointed out, after knocking down LRP3, the number of chondrocytes was reduced compared to the control group. Subsequently, we further performed CCK-8 analysis, and the results showed that the viability and growth of chondrocytes were inhibited after LRP3 knockdown (The following figure, data is not shown in the text). In the subsequent GAG content analysis, in order to avoid interference caused by different cell numbers, we also detected the DNA content of the samples, and used the form of GAG/DNA in the final data presentation to normalize the result (Fig.4e, j).

8. Comment:

Possible function of LRP3 should be discussed. It is not clear at all about how is LRP3 involved in Ras signaling pathway or SDC4 expression.

Response:

Thank you for your suggestion. To explore the mechanism of LRP3 in OA, we performed RNA sequencing (RNA-seq) analysis. According to the results of RNA-seq analysis, we found that after knocking down LRP3, the expression of SDC4 was significantly increased, and the Ras signaling pathway was activated at the same time. Our subsequent experiments also verified the sequencing results. As a member of the LRP family, LRP3 is one of the evolutionary conserved cell surface receptors. Previous studies have shown that members of the LRP family have a variety of biological functions, including cell signal transduction, synaptic plasticity regulation and cell fate determination. The results we found through RNA-seq analysis may be an unreported function of LRP3, and the specific mechanism behind it needs further in-depth study. Our research group will also be committed to continuous research in this field.

Reviewer #3:**1. Comment:**

Major conceptual issue is the seriously misleading statement (L54) that “hypercholesterolemia has been proved to be an independent risk factor for systemic OA.” A reference from 2003 supporting this claim is cited: “Al-Arfaj AS. Radiographic osteoarthritis and serum cholesterol. Saudi Med J 24, 745-747 695 (2003). “In contrast, the recent (Sep 22, 2020) data of Schwager et al (doi: 10.1002/acr.24455.) is neither cited here nor discussed. It studied participants from the Multicenter Osteoarthritis study (MOST) cohort at risk of developing knee OA from baseline through 7 years. The data revealed that neither total cholesterol, LDL nor HDL showed a significant association with radiographic or symptomatic OA. Additionally, they found no association of these lipid measures with cartilage loss, worsening synovitis or worsening knee pain, rejecting a possible association between total cholesterol, LDL or HDL with OA outcomes. As this is the clinical basis of the work, this ignorance of serious published work is deeply troubling.

Response:

Thanks for your professional advice. Our previous expression was indeed not rigorous enough,

which caused unnecessary misunderstandings. Therefore, we have revised the corresponding Introduction and Discussion sections according to your suggestion, citing the latest literature, and discussing the current controversies of the relationship between cholesterol and OA.

Modifications of the Introduction section:

“Despite high prevalence and societal impact, the aetiology of OA remains unclear. OA is a multifactorial disease with many possible causes including aging², trauma³, sex⁴, abnormal biomechanics⁵ and obesity⁶. Furthermore, new evidence supports a link between OA and metabolic syndrome (MetS), such as hypertension, insulin resistance and dyslipidemia, which are associated with an increased risk of OA⁷. In the past few years, many animal studies have been conducted to reveal the role of cholesterol in the pathogenesis of OA^{8, 9, 10, 11}. These findings indicated that different OA animal models fed with high-cholesterol diet led to increased cartilage damage, elevated adiposity-derived inflammation and increased OA severity. In the field of clinical research, several lines of evidence have demonstrated that some OA patients have significantly higher serum levels of low-density lipoprotein (LDL) compared to healthy controls^{12, 13}. Al-Arfaj AS’ research showed an association between high serum cholesterol level and both knee and generalized OA, proving that hypercholesterolemia may be an independent risk factor for systemic OA¹⁴. Meanwhile, a cross-sectional study in Korea showed that high cholesterol was positively correlated with OA pain¹⁵. Above evidences indicated it is highly likely that there is a metabolic OA phenotype which is associated with high cholesterol levels.

In addition, anomalous homeostasis of cholesterol is also shown to be one of the characteristic features of OA development, with accumulation of cholesterol found particularly in the superficial zone of the cartilage¹⁶. Functional genomics analysis of osteoarthritic cartilage and chondrocytes added stronger evidence supporting the role of cholesterol-related genes in OA pathogenesis⁷. Wan-Su Choi et al.’s research on Nature indicated that OA is a disease associated with metabolic disorders and suggest that targeting the CH25H (cholesterol 25-hydroxylase)-CYP7B1(25-hydroxycholesterol 7 α -hydroxylase)-ROR α axis of cholesterol metabolism may provide a therapeutic avenue for treating osteoarthritis¹⁰. Based on existing clinical and basic research, cholesterol metabolism and its related genes may be involved in the occurrence and development of OA, although the relationship between high cholesterol and the OA is still controversial in the field of clinical cohort research^{17, 18, 19}. However, the underlying molecular mechanism of

cholesterol metabolism and its related genes involved in OA pathogenesis still unclear and needs to be explored²⁰.”

Modifications of the Discussion section:

“OA is the most prevalent chronic joint disease mainly affects individuals over the age of 65³⁴. Although several risk factors related to OA have been pointed out, including aging, gender, muscle strength, obesity and mechanical alignment¹, the pathogenesis of OA remains largely unclear. MetS is a complex group of metabolic disorder syndromes, including a number of conditions, like abdominal obesity, impaired fasting glucose, hypertension (HTN) and dyslipidaemia (high cholesterol and low high-density lipoprotein)³⁵. Since both of MetS and OA pose significant challenges to public health, a large number of studies have been conducted to investigate the association between them. More specifically, Vasiliki Gkretsi et al. reported a positive correlation between increased cholesterol levels and knee, hip, hand and/or generalized OA through epidemiological studies³⁶. A cohort study on knee OA showed that hyperlipidemia is associated with joint pain and an increased prevalence of OA³⁷. The pain and OA severity scores of OA patients with hypercholesterolemia were also higher than those of the control group³⁸. In addition, Kellgren et al. first showed an association between elevated serum cholesterol levels and hand OA in women³⁹. At the same time, a study involving 1003 middle-aged women revealed that elevated serum total cholesterol levels were associated with unilateral and bilateral knee OA⁴⁰. In this study, we also found that serum TC and LDL in some OA patients were significantly higher than that of non-OA patients, which verified the positive correlation between clinical hyperlipidemia and OA. As for basic research, both global apolipoprotein E knockout (ApoE^{-/-}) mice and rats fed with HCD were proved to have severity of OA phenotype^{10,41}, which was consistent with our study. In addition, we used exogenous cholesterol to stimulate human cartilage explants in vitro, and the results showed a significant proteoglycans and type II collagen reduction. Similarly, Saba Farnaghi et al. also found the same change trend when stimulating bovine cartilage explants with cholesterol⁴¹. Furthermore, studies showed that administration of atorvastatin can reduce the serum cholesterol level while effectively delaying the progression of OA⁹. Although the above results showed a positive correlation between cholesterol levels and OA, a number of studies have reported contradictory results. The data of Schwager et al. ’s research revealed that neither total cholesterol, LDL nor HDL

showed a significant association with radiographic or symptomatic OA, as well as cartilage loss, worsening synovitis or worsening knee pain¹⁷. In addition, in the Rotterdam study total/HDL cholesterol ratio was not associated with overweight and hand OA in 3585 participants over 55 years¹⁹. The differences in study results may be caused by differences in the number of people enrolled in the study cohort, the diagnostic criteria for OA, or the geographical distribution of the population. Although the relationship between high cholesterol and the occurrence and development of OA is still controversial in the clinical cohort study, based on existing clinical and basic research, it is highly likely that there is a metabolic OA phenotype that is associated with high cholesterol levels. However, the mechanisms underlying the complex interactions between lipid metabolism, especially cholesterol metabolism, and OA has not yet been completely elucidated⁴².”

2. Comment:

Logical issue #1: Figure 1 E, OARSI grade suggests that a high cholesterol diet already induces OA without any additional ACLT, shown by the significant differences in MFC and MTP. From a clinical point of view this would mean that patients with a high cholesterol diet would have an increased incidence of OA, which is not supported by clinical studies. Moreover, the data also suggest that following ACLT, structural or a progression is worse in high cholesterol diet animals, again, a finding that is not supported by clinical data.

Response:

Thank you for pointing out this important issue. In response to this problem, first of all, the results of our research group's animal experiments are consistent with those of many published studies (PMID: 30728500, 27737897). In addition, a small number of clinical studies have shown that dietary-induced metabolic syndrome may be associated with the risk of OA (PMID: 32399062). Finally, the focus of this study is to elicit the possible important roles of cholesterol metabolism-related genes in the pathogenesis of OA through the relationship between cholesterol and OA. Of course, our research group will continue to explore in clinical research in this field.

3. Comment:

Logical issue #2: If the 20 patients (Supplementary Figure 1) with OA have a significantly higher BMI, then the BMI-induced overload could play a role in OA, not the associated hyperlipidemia.

Response:

Thanks for pointing out this question. First of all, based on your suggestions, we re-collected the data, increased the number of OA patients to 100, and at the same time increased the number of control groups to 91, and re-analyzed the data. Our results showed that the TC of OA patients was higher than that of the control group, but such observational studies cannot completely deny the role of BMI. Existing clinical studies also have such limitations because the two factors of BMI and hyperlipidemia are closely related. This is also the reason why existing clinical studies are controversial on whether lipid metabolism affects OA. Our research is mainly to further explore the possible role of cholesterol metabolism-related genes in OA by studying the relationship between cholesterol and OA. OA is a systemic multifactorial disease, if a certain factor needs to be clarified, it is necessary to further search for some people with abnormal lipid metabolism caused by lipid metabolism gene mutations, but no changes in BMI for research.

4. Comment:

Major technical issue #1: P4 L91 “which indicated that the hyperlipidemia was significantly positively correlated with the development of OA (Fig. 1a).” No correlation analysis was shown on Fig1a or in the whole study, so there is no evidence for this statement. This reviewer would expect from authors submitting such a sophisticated analysis to know the difference between an “association” and true “correlation”.

Response:

Thank you for your suggestion. In order to describe the result more accurately, we changed it to: “We found that compared with non-OA patients, TC and LDL were significantly higher in OA patients, accompanied by larger BMI and more metabolic diseases, which indicated that there may be a link between hyperlipidemia and the development of OA (Fig. 1a).”

5. Comment:

Major technical issue #2: Proteoglycans and type II collagen are measured by IOD value. True biochemical quantification is however standard and needed (e.g., DMMB).

Response:

For the analysis of immunohistochemical staining results, we used ImageJ software to analyze

the gray value of the effective staining area, and calculated the corresponding integrated optical density (IOD) for semi-quantitative analysis and comparison. This traditional immunohistochemical semi-quantitative analysis method has been applied in the published literature (PMID: 31015473, 33550074, 34611550). Although it cannot be accurately quantified, it has been widely used for comparative analysis with the control group. Moreover, we used the DMMB method to analyze and compare the proteoglycan content of chondrocytes in *in vitro* experiments.

6. Comment:

Major technical issue #3: The Western blots shown in the Supplementary materials in larger size are inadequate to assess the reliability of the performed experiments.

Response:

Thank you for your professional advice. We have repeated all the western blots results of this experiment and provided a new uncropped and original full film figures with indicate molecular weight markers in the supplementary data, making the results clearer and more credible.

7. Comment:

Major technical issue #4: The statistical tests are inadequate. For comparing non-parametric data, e.g., OARSI grades (Fig1f, Fig2f, j, Fig3l, p, Fig4c), non-parametric tests (e.g., Mann-Whitney rank sum test, Kruskal-Wallis ANOVA) are needed. In the other data were there any cases when the normality test failed? If yes, please describe which tests were used for non-normally distributed data.

Response:

Thank you for pointing this out, it was an oversight in our description of the statistical method. Following your suggestion, we have revised the description in the Statistical Methods section to read: “All statistical analyses were performed using SPSS (version 20.0; IBM Corp, Chicago, IL, USA). Data are presented as mean \pm standard deviation. The Shapiro–Wilk test was used for testing data normality, and the Levene’s test was used for testing homogeneity of variance. The student’s t-test (parametric data) or Mann-Whitney rank sum test (for non-parametric data) were performed to compare two independent groups. For comparison between multiple values, one-way analysis of variance (ANOVA) (parametric data) or Kruskal-Wallis test (for non-parametric data) with Tukey’s post hoc test to assess differences between specific groups. n indicates the number of

biologically independent samples. Significant data are indicated by * $P < 0.05$; ** $P < 0.01$; and *** $P < 0.001$.”

8. Comment:

The Western blots shown in the Supplementary materials in larger size are still inadequate to assess the reliability of the performed experiments. In Supplementary materials please show the whole membranes of the Western blots including the whole protein ladder on each. Please also show 3 biological or technical repeats of each experiment. Finally, and most importantly, please show the amido black staining of the entire membranes for each experiment to be able to compare the whole protein content of the loaded samples.

Response:

Thank you for your professional advice. We have repeated all the western blots results of this experiment and provided a new uncropped and original full film figures with indicate molecular weight markers in the supplementary data. In our Western blots experiment, in order to ensure that the total amount of protein used when loading the protein sample is consistent, and to eliminate the experimental error in the loading process, we adopted the internal control method. We selected the housekeeping protein GAPDH, which is stable and highly expressed in similar cells, as an internal control. All of our sample loading operations for WB experiments are performed based on the same expression of the loaded protein sample GAPDH.

9. Comment:

P2 L51: “Previous studies largely believed that OA cartilage degeneration is related to aging, trauma, and abnormal biomechanics. The existing more evidence shows that metabolic syndrome is associated with an increased risk of OA, especially lipid metabolism” This sentence is phrased so that the reader might think that these previous studies are not true or outdated, and OA is only linked to metabolic syndrome. Please rephrase the sentence and emphasize that OA is a multifactorial disease with many possible causes.

Moreover, as described in references 26-27, neither metabolic syndrome nor its components, with the exception of hypertension, are associated with incident OA after adjustment for BMI or body weight.

Response:

Sorry for the misunderstanding in our description, we revised it to: “OA is a multifactorial disease with many possible causes including aging², trauma³, sex⁴, abnormal biomechanics⁵ and obesity⁶. Furthermore, new evidence supports a link between OA and metabolic syndrome (MetS), such as hypertension, insulin resistance and dyslipidemia, which are associated with an increased risk of OA⁷.”

As for references 26-27, what we described here was not rigorous enough, so we modified it in the Discussion section: “OA is the most prevalent chronic joint disease mainly affects individuals over the age of 65³⁴. Although several risk factors related to OA have been pointed out, including aging, gender, muscle strength, obesity and mechanical alignment¹, the pathogenesis of OA remains largely unclear. MetS is a complex group of metabolic disorder syndromes, including a number of conditions, like abdominal obesity, impaired fasting glucose, hypertension (HTN) and dyslipidaemia (high cholesterol and low high-density lipoprotein)³⁵. Since both of MetS and OA pose significant challenges to public health, a large number of studies have been conducted to investigate the association between them. More specifically, Vasiliki Gkretsi et al. reported a positive correlation between increased cholesterol levels and knee, hip, hand and/or generalized OA through epidemiological studies³⁶. A cohort study on knee OA showed that hyperlipidemia is associated with joint pain and an increased prevalence of OA³⁷. The pain and OA severity scores of OA patients with hypercholesterolemia were also higher than those of the control group³⁸. In addition, Kellgren et al. first showed an association between elevated serum cholesterol levels and hand OA in women³⁹. At the same time, a study involving 1003 middle-aged women revealed that elevated serum total cholesterol levels were associated with unilateral and bilateral knee OA⁴⁰. In this study, we also found that serum TC and LDL in some OA patients were significantly higher than that of non-OA patients, which verified the positive correlation between clinical hyperlipidemia and OA. As for basic research, both global apolipoprotein E knockout (ApoE^{-/-}) mice and rats fed with HCD were proved to have severity of OA phenotype^{10,41}, which was consistent with our study. In addition, we used exogenous cholesterol to stimulate human cartilage explants in vitro, and the results showed a significant proteoglycans and type II collagen reduction. Similarly, Saba Farnaghi et al. also found the same change trend when stimulating bovine cartilage explants with cholesterol⁴¹.

Furthermore, studies showed that administration of atorvastatin can reduce the serum cholesterol level while effectively delaying the progression of OA⁹. Although the above results showed a positive correlation between cholesterol levels and OA, a number of studies have reported contradictory results. The data of Schwager et al. 's research revealed that neither total cholesterol, LDL nor HDL showed a significant association with radiographic or symptomatic OA, as well as cartilage loss, worsening synovitis or worsening knee pain¹⁷. In addition, in the Rotterdam study total/HDL cholesterol ratio was not associated with overweight and hand OA in 3585 participants over 55 years¹⁹. The differences in study results may be caused by differences in the number of people enrolled in the study cohort, the diagnostic criteria for OA, or the geographical distribution of the population. Although the relationship between high cholesterol and the occurrence and development of OA is still controversial in the clinical cohort study, based on existing clinical and basic research, it is highly likely that there is a metabolic OA phenotype that is associated with high cholesterol levels. However, the mechanisms underlying the complex interactions between lipid metabolism, especially cholesterol metabolism, and OA has not yet been completely elucidated⁴².”

10. Comment:

P2 L51 Please also mention that obesity (or high BMI), which is a well-known risk factor of OA by increasing joint load, is strongly connected to metabolic syndrome.

Response:

Thanks for your suggestion, we revised it to: “OA is a multifactorial disease with many possible causes including aging, trauma, sex, abnormal biomechanics and obesity.”

11. Comment:

P2 L52 “The existing more evidence”. Please rephrase.

Response:

Thanks for your suggestion, we revised it to: “Furthermore, new evidence supports a link between OA and metabolic syndrome (MetS), such as hypertension, insulin resistance and dyslipidemia, which are associated with an increased risk of OA.”

12. Comment:

P3 L74 “Aggrecanases belong to the ADAMTS family of metalloproteinases, and the degradation of aggrecan by ADAMTS-4 and ADAMTS-5 20.” Something is missing from this sentence.

Response:

Thanks for your suggestion, we revised it to: “Aggrecanases belong to the ADAMTS family of metalloproteinases, and the main ones that degrade aggrecans are ADAMTS-4 and ADAMTS-5.”

13. Comment:

P3 L78 “Here, we investigated the functional role of the cholesterol-LRP3-SDC4 axis in OA development in this study.” “Here” or “in this study” is redundant.

Response:

Thanks for your suggestion, we revised it to: “Here, we investigated the functional role of the cholesterol-LRP3-SDC4 axis in OA development.”

14. Comment:

P3 L80 “The elucidation of such mechanisms could facilitate the development of new and effective therapeutic targets for the treatment of OA.” Instead of “therapeutic targets” therapies would fit better.

Response:

Thanks for your suggestion, we revised it to: “The elucidation of such mechanisms could facilitate the development of new and effective therapies for the treatment of OA.”

15. Comment:

The BMI data are repeated in Fig1a and Supp Fig1C and the table of Supp Fig1a-b. Please present them only once.

Response:

Thanks for your suggestion, we revised the figure and only showed BMI data in revised Fig1a.

16. Comment:

P4 L91 “which indicated that the hyperlipidemia was significantly positively correlated with the

development of OA (Fig. 1a).” No correlation analysis was shown on Fig1a or in the whole study, so there is no evidence for this statement.

Response:

Thank you for your suggestion. This is indeed an error in our statement. In order to describe the result more accurately, we changed it to: “We found that compared with non-OA patients, TC and LDL were significantly higher in OA patients, accompanied by larger BMI and more metabolic diseases, which indicated that there may be a link between hyperlipidemia and the development of OA (Fig. 1a).”

17. Comment:

P4 L93 “Safranin-O and toluidine blue staining showed a significant proteoglycans reduction in response to cholesterol and TNF- α ”. Why 50 $\mu\text{g}/\text{mL}$ cholesterol concentration was applied in vitro, if the normal level in the blood is $<200 \text{ mg}/\text{dl}$ ($<2000 \mu\text{g}/\text{ml}$), and pathological high total cholesterol level starts from $>240 \text{ mg}/\text{dl}$ ($>2400 \mu\text{g}/\text{ml}$)? Is cholesterol present in the synovial fluid in obese people? If yes, in what concentration?

Response:

Human serum total cholesterol includes free cholesterol and cholesterol esters, of which free cholesterol accounts for about one-third. The total cholesterol in the serum does not directly act on the cartilage, the real effect is the free cholesterol in the synovial fluid. It has been reported in the literature that free cholesterol is basically undetectable in the synovial fluid of normal people, while the concentration of total cholesterol in the synovial fluid of OA patients is about $700 \mu\text{g}/\text{mL}$ (PMID: 33870758). We used the water-soluble cholesterol (sigma) in this paper, and literatures recommended using $30 \mu\text{g}/\text{mL}$ to directly stimulate chondrocytes (PMID: PMID: 27737897). Through experiments, the highest solubility of this water-soluble cholesterol is $50 \mu\text{g}/\text{mL}$, otherwise it will precipitate in water. Although there is still a certain gap between the concentration we used and $700 \mu\text{g}/\text{mL}$ reported in the literature, it can simulate a high cholesterol environment in the joint cavity.

18. Comment:

P4 L105 “HCD prominently down-regulated the chondrocytes anabolic genes (Col2a1, Acan and

Sox9), and upregulated the anabolism gene (Mmp13 and Adamts5) (Fig. 1g).” Sox9 is not shown on the figure. Mmp13 and Adamts5 are catabolism genes.

Response:

We have corrected the typo in the corresponding section of the text and modified it to: “**RNA sequencing (RNA-seq) analysis indicated that HCD prominently down-regulated the chondrocytes anabolic genes (*Col2a1*, *Acan* and *Sox9*), and upregulated the catabolism genes (*Mmp13* and *Adamts5*) (Fig. 1g).**” At the same time, we reconstructed the sequenced heatmap according to your suggestion and displayed it in the new figure 1g.

19. Comment:

P4 L118 “from 5 trauma patient (as control)”. Patients

Response:

We have corrected the typo in the corresponding section of the text as you suggested.

20. Comment:

P4 L120 “the surface of OA cartilage displayed severely degenerated and exhibited cracks and fissures,” something is missing from the sentence

Response:

We modified it to: “**Safranin-O and toluidine blue staining showed that compared with normal cartilage, the staining intensity of cartilage tissue in OA patients was significantly lower (Fig. 3a, b).**”

21. Comment:

P5 L126 “Senile Mice were mainly manifested by the”. Strange phrasing

Response:

We modified it to: “**Similar observations were observed in the senescence-induced mouse OA model. The articular cartilage of 12-month-old mice was more severely degenerated than that of 3-month-old mice, and the content of proteoglycans was significantly reduced (Fig. 3i, j), while LRP3 protein was also significantly reduced (Fig. 3k, l).**”

22. Comment:

In Fig2m and Fig3a, f the expression of Col2, Acan, Sox9 and LRP3 is shown separately in rat chondrocytes, and TNF- α induced rat OA chondrocytes with and without LRP3 knockdown or overexpression. A combination image of the absolute values of Fig2m and Fig3a, f could also be interesting (maybe to be shown as a supplementary figure) to see the differences between the control groups and the treatment groups, and to see whether the overexpression of LRP3 in OA chondrocytes restore the levels of Col2, Acan, Sox9 and LRP3 to the normal levels.

Response:

This is a good suggestion. We re-performed a unified horizontal comparison of normal chondrocytes, TNF- α induced rat OA chondrocytes and LRP3 overexpressed OA chondrocytes by RT-PCR and WB experiments. The results showed that overexpression of LRP3 in TNF- α -induced rat OA chondrocytes significantly increased the expression of COL2A1, ACAN and SOX9 to near normal levels. The corresponding results were shown in the revised figure 4f, g.

23. Comment:

In Supp Fig S5 it should be mentioned that vehicle is Lv-con313

Response:

We have made modifications in the corresponding figures.

24. Comment:

P6 L162-164: Supp Fig S7 should be cited between S6 and S8. In Supp Fig S7 the body size of only the newborn mice is shown. Is there any difference in adult mice between age-matched WT and LRP3^{-/-}? If LRP3 mediates lipid metabolism, it might affect the adult body weight too. Also, do LRP3-KO mice gain weight in response to high-cholesterol diet similarly to WT mice shown in Fig1e?

Response:

We re-compared adult WT and *Lrp3*^{-/-} mice as you suggested and the corresponding results were presented in the new supplementary Fig. S5: “We further compared the difference between *Lrp3*^{-/-} mice and wild type (WT) mice in physical development. The results showed that there was no obvious disability and dysplasia in *Lrp3*^{-/-} mice (supplementary Fig. S5a). Under regular diet

(RD), adult (3 months) *Lrp3*^{-/-} and WT mice had no significant difference in body weight (supplementary Fig. S5b). However, in the case of high-cholesterol diet (HCD), the weight of the WT mice was significantly greater than the weight of *Lrp3*^{-/-} mice (supplementary Fig. S5c). Similar to WT mice, *Lrp3*^{-/-} mice also gained weight (supplementary Fig. S5d) and TC in response to HCD(supplementary Fig. S5e).”

25. Comment:

Supp Fig S8b: the explanation for the red-blue colors is missing

Response:

It was an oversight on our part, we added the definition of the red-blue color pillars in the corresponding picture.

26. Comment:

P7 and everywhere else in the manuscript: All figures, figure panels and tables should be presented in order. For example, the description of panel A of figure 3 should not come before that of panel B of figure 2. Now it is Fig4a, b, f, b, d, c, e, f; also, on page 8 for Fig 5.

Response:

Thank you for pointing out the problem with the ordering of the figure panels in our article. We reviewed the full text and elicited the figure panels in the correct order.

27. Comment:

P7 L206 “OARSI scores was” - scores were or score was

Response:

We have corrected the typo in the corresponding section of the text as you suggested.

28. Comment:

P8 L217 “up-regulated in responded to Lrp3 knockdown” - in response to

Response:

We have corrected the typo in the corresponding section of the text as you suggested.

29. Comment:

What was the specific reason to choose SDC4 for further analysis over the many other possible candidates which also showed altered expression?

Response:

We screened a large number of candidate genes through RNA-sequencing. Among these candidate genes, we first sorted the differential genes, and extracted the top 20 genes with large differences in expression. Secondly, the genes with extremely low basal expression in chondrocyte were excluded. Then by reviewing the literature, we selected genes reported to be associated with OA and cartilage degeneration. Through three rounds of screening, SDC4 appeared to rank first among the remaining candidate genes, so we finally selected it for further analysis.

30. Comment:

P8 L228 and everywhere else: KEGG, MEK, ERK, etc.: please define the abbreviations where they are mentioned first.

Response:

Based on your suggestion, we have added definitions of abbreviations to the corresponding sections of the text.

31. Comment:

P9 L249, 258 typo “western bolt”

Response:

We have corrected the typo in the corresponding section of the text as you suggested.

32. Comment:

P10 L281 “In this paper, we have made several key observations presented here.” Redundant “In this paper” or “presented here.”

Response:

We have deleted the “presented here” in the corresponding place as you suggested.

33. Comment:

P10 L289 "... chondrocytes. And the overexpression" Generally, do not start a sentence with "and".

Response:

We have deleted the "and" in the corresponding place as you suggested.

34. Comment:

P10 L296 "In recent years, a large number of epidemiological and basic researches have shown that metabolic syndrome, which is dominated by abnormal cholesterol metabolism, is closely associated with the occurrence and development of OA^{26, 27.}" In fact, both references state the opposite, that neither metabolic syndrome nor its components, with the exception of hypertension, are associated with incident OA after adjustment for BMI or body weight.

Response:

What we described here was not rigorous enough, so we modified it in the Discussion section: "OA is the most prevalent chronic joint disease mainly affects individuals over the age of 65³⁴. Although several risk factors related to OA have been pointed out, including aging, gender, muscle strength, obesity and mechanical alignment¹, the pathogenesis of OA remains largely unclear. MetS is a complex group of metabolic disorder syndromes, including a number of conditions, like abdominal obesity, impaired fasting glucose, hypertension (HTN) and dyslipidaemia (high cholesterol and low high-density lipoprotein)³⁵. Since both of MetS and OA pose significant challenges to public health, a large number of studies have been conducted to investigate the association between them. More specifically, Vasiliki Gkretsi et al. reported a positive correlation between increased cholesterol levels and knee, hip, hand and/or generalized OA through epidemiological studies³⁶. A cohort study on knee OA showed that hyperlipidemia is associated with joint pain and an increased prevalence of OA³⁷. The pain and OA severity scores of OA patients with hypercholesterolemia were also higher than those of the control group³⁸. In addition, Kellgren et al. first showed an association between elevated serum cholesterol levels and hand OA in women³⁹. At the same time, a study involving 1003 middle-aged women revealed that elevated serum total cholesterol levels were associated with unilateral and bilateral knee OA⁴⁰. In this study, we also found that serum TC and LDL in some OA patients were significantly higher than that of non-OA patients, which verified the positive correlation between clinical hyperlipidemia and OA.

As for basic research, both global apolipoprotein E knockout (ApoE^{-/-}) mice and rats fed with HCD were proved to have severity of OA phenotype^{10, 41}, which was consistent with our study. In addition, we used exogenous cholesterol to stimulate human cartilage explants in vitro, and the results showed a significant proteoglycans and type II collagen reduction. Similarly, Saba Farnaghi et al. also found the same change trend when stimulating bovine cartilage explants with cholesterol⁴¹. Furthermore, studies showed that administration of atorvastatin can reduce the serum cholesterol level while effectively delaying the progression of OA⁹. Although the above results showed a positive correlation between cholesterol levels and OA, a number of studies have reported contradictory results. The data of Schwager et al. 's research revealed that neither total cholesterol, LDL nor HDL showed a significant association with radiographic or symptomatic OA, as well as cartilage loss, worsening synovitis or worsening knee pain¹⁷. In addition, in the Rotterdam study total/HDL cholesterol ratio was not associated with overweight and hand OA in 3585 participants over 55 years¹⁹. The differences in study results may be caused by differences in the number of people enrolled in the study cohort, the diagnostic criteria for OA, or the geographical distribution of the population. Although the relationship between high cholesterol and the occurrence and development of OA is still controversial in the clinical cohort study, based on existing clinical and basic research, it is highly likely that there is a metabolic OA phenotype that is associated with high cholesterol levels. However, the mechanisms underlying the complex interactions between lipid metabolism, especially cholesterol metabolism, and OA has not yet been completely elucidated⁴².”

35. Comment:

P11 L303 please define “ApoE^{-/-} mice”

Response:

We revised it to: “global apolipoprotein E knockout (ApoE^{-/-}) mice.”

36. Comment:

P11 L312 “However, the molecular mechanism of cholesterol-induced OA cartilage degeneration has not yet been completely eliminated.” - Elucidated

Response:

We have revised this sentence from the manuscript: “**However, the mechanisms underlying the**

complex interactions between lipid metabolism, especially cholesterol metabolism, and OA has not yet been completely elucidated.”

37. Comment:

P12 L359 “we identified that SDC4 as the downstream molecular target”. “That” is not needed in the sentence

Response: We removed “that” from this sentence as you suggested.

38. Comment:

In Fig2, human control and OA histological sections were shown for LRP3 expression. Can the authors show the expression of SDC4 also in these samples?

Response:

We provided immunohistochemical staining for SDC4 on these same samples and presented them in the revised Figure 7a, b, c.

39. Comment:

P13 L377-381 The patient numbers are not matching, please clarify them: were the n=40 patients which were evaluated for clinical data also used for cartilage samples?

“OA n=20, 14 females and 7 males” their sum is 21, not 20.

Response:

Thank you for pointing out our mistake here. The actual number was 21 instead of 20.

40. Comment:

P13 L388 “anterior cruciate ligament transplantation (ACLT)” - transection

Response: We have corrected the spelling in the corresponding position of the manuscript.

41. Comment:

P14 L403: n=? of the rats is missing

Response: In total, we extracted chondrocytes from 35 rats. We have also added "n=35" at the corresponding position in the manuscript.

42. Comment:

P16 L449, P17 L470 typo: “VAGSVIL” instead of ARGSVIL

Response: We have revised “VAGSVIL” to “ARGSVIL” according to your suggestion.

43. Comment:

P17 L464 please add reference for OARSI grading

Response: We have added the corresponding references as you suggested.

44. Comment:

P22 L633 “21□, and then incubated with Filipin Staining solution for 30 min at 21□ in the dark.”

Some characters are not displayed correctly.

Response: It should be 21°C here, we have corrected the character in the manuscript.

45. Comment:

Fig1c typo: collogen

Response: We corrected it to “collagen”.

46. Comment:

Fig1l typo: humam

Response:

Sorry for the typo. We have corrected the spelling in the corresponding figure.

47. Comment:

Fig 1l: What comparisons do the significance signs show? Are they compared to the red column (control).?

Response:

Yes, the significance signs here indicated the statistical difference between each column and the red column (control).

48. Comment:

Fig2b: the two diagrams are the same. In the raw data excel file the original datasets are different, so it is probably only a mistake of inadvertence.

Response:

Thank you for pointing out our mistake here. We replaced the correct diagrams in their corresponding positions.

49. Comment:

Fig2e: The staining of the bone and the marrow looks also faint on the ACLT image, not only that of the cartilage. Were they made with the same slice thickness, staining protocol and imaging settings? Please provide comparable images.

Response:

Sorry for the misunderstanding here. All our paraffin sections were of uniform thickness. The staining procedure and imaging settings were also consistent. We have provided comparable images in the revised figure as you suggested.

50. Comment:

Fig 2g: Why is the bone marrow less stained in ACLT? Were the images made with the same slice thickness, staining protocol and imaging settings? If not, please provide comparable images.

Response:

Sorry for the misunderstanding here. All our paraffin sections were of uniform thickness. The staining procedure and imaging settings were also consistent. We have provided comparable images in the revised figure as you suggested.

51. Comment:

Fig3b, g: Why is LRP3 so bad quality and pixelated in these panels while on all other Western blots they look fine?

Response:

Thank you for bringing up our flaw. This is due to the heterogeneity of image quality caused by different batches of WB experiments. In order to improve the quality, we purchased a new LRP3

antibody (PA5-50051; Invitrogen, USA) and repeated the western blots experiment. We also provided new uncut film figures with molecular weight markers to make the results clearer and more convincing.

52. Comment:

Fig3k: Scale values are missing in some panels while on some others they are redundant.

Response:

We have added and modified the corresponding panels' scale values according to your suggestion.

53. Comment:

Fig4c, d,e: typo: Hige

Response:

Thank you for pointing out our spelling mistake, we corrected it to "High".

54. Comment:

Fig4e: "E" is capital

Response:

We corrected it to the lowercase letter "e".

55. Comment:

Fig4g: Why are the menisci missing in vehicle and low dose? In these two groups, especially in vehicle, the subchondral bone looks extremely injured. What is the explanation of this?

Response:

Thanks for your suggestion. Sorry for the misunderstanding caused by our CT slice selection. None of the experimental groups had the menisci removed nor subchondral bone destroyed. In order to show the results more clearly, we re-selected the CT slices and re-presented the results with coronal CT slices (revised Fig. 5g).

56. Comment:

Please describe the selection of the subarticular spongiosa ROIs for micro-CT measurements in more details in the Methods (how were the boundaries chosen, was it an automated algorithm or manual drawing, etc.?).

Response:

Thanks to your suggestion, we have added a more detailed description in the corresponding Materials and Methods section: “Histomorphometric analysis was performed on longitudinal images of the femur subchondral bone, which was also analysed using the Inveon Research Workplace software (Siemens Inveon MM Gantry, Berlin, Germany), determining the trabecular ratio of bone volume to tissue volume (BV/TV) and trabecular bone pattern factor (Tb. Pf.). Briefly, we created 3D analyses of the trabecular bone. The region of interest (ROI) of the trabecular bone was drawn beginning from 5% of the femur length proximal to the distal metaphyseal growth plate and extending proximally for another 5% of the total femur length. All the ROI was segmented manually on a slice-by-slice basis. The BV/TV and Tb. Pf. were collected from the 3D analyses data and used to represent the trabecular bone parameters.”

57. Comment:

Fig5f: what is the difference between the green and orange columns? Under both Lv-RLP3 is written.

Response:

We are sorry that the description in figure caused misunderstanding. The green and orange columns were indeed the experimental groups injected with Lv-LRP3. The difference is that the green column represented the low-dose injection group, and the orange column represented the high-dose injection group. In order to express more clearly, we have modified the corresponding pictures and marked different dosages.

58. Comment:

Fig5g: typo: KRGG

Response:

Thank you for bringing up our low-level spelling error, we corrected it to KEGG.

REVIEWER COMMENTS

Reviewer #1 (Remarks to the Author):

Thank you for your responses to my comments and the other reviewers. I have carefully examined your responses to my comments and I am satisfied that you have satisfactorily addressed my concerns. You submitted a clean version of the revised manuscript. This version did not have any track changes so it was difficult too see exactly where in the text the changes had been made (although you included some of the changes in your responses to reviewers document).

Reviewer #4 (Remarks to the Author):

The authors addressed the concerns of the previous referee. However, the three reviewers raised the suspicious problem of LRP3 antibody western blot results, and the author solved the problem by changing and purchasing a new antibody and repeating the test, which itself increased the suspicious nature of the data. Considering the whole project, LRP3 runs through the whole process, including cell growth and protein degradation in animal cells. If the important role of LRP3 is to be explained, and if the antibody is replaced, the GAPDH band should also be replaced. That is, if the internal reference is replaced, other antibody bands will also be changed, and other genes cannot use the previous antibody bands.

Besides, there are many picture errors and inconsistent data, resulting in doubts about the authenticity of the conclusion.

1, Figure 1b, in group of "TNF-a+Cholesterol", the pathological staining samples were from different specimens, especially for the COLII staining. The surface of the cartilage in COLII staining were intact, while the cartilage in Safranin O staining and Toluidine blue were seriously damaged. This is unreasonable.

2, Figure 3i and 3k, there are also cases where the pictures are not consistent. The mouse knee cartilage of 12m in Figure 3i were damaged, while the cartilage in Figure 3k were relatively normal, why are the samples different in the same period?

3, Figure 4k, the LRP staining indicated that both nucleus and cytoplasm were stained, while in figure s3a, the LRP staining were in nucleus, what is the specific location of the LRP3 staining in the cell? The location and intensity of cell staining are different. Please confirm whether the picture is misplaced.

4, Figure 6i, according to significance mark on the left, Ras signaling pathway were marked blue. According to the label, there should be no statistical significance for this pathway.

5, Figure s1b, the staining for COL2A1 and LRP3 were overexposed, which can be seen from DAPI. In addition, the expression of LRP3 was inconsistent in the upper and lower pictures. The upper picture showed that it was expressed in the nucleus and the next one was not. It is suspected of manual brightness adjustment, resulting in unreliable data.

6, Figure s3b, LV silrp3 sham group safranin O staining showed cartilage surface were damaged, while LRP3 group showed intact cartilage surface and different cell number and arrangement of chondrocytes. The data were not accurate enough. Similarly, in the lv-silrp3 ACLT group, the cartilage samples of safranin O staining and the samples of LRP3 and col2 are not the same. The unreliable data resulted in the weakening of the authenticity of the conclusion.

7, Figure s6a, the LRP-/- ACLT group cartilage safranin O staining was seriously damaged, while the

cartilage surface in LRP3 group and col2 group was intact. The pictures were selected from different periods?

8, Figure s8b, It was obvious that the staining of col4-sv group was not the same as that of col4-sv group in the same cartilage sample. Please explain why the staining of col4-sv group was not the same as that of col4-sv group in the same sample.

Reviewer #1:

1. Comment:

Thank you for your responses to my comments and the other reviewers. I have carefully examined your responses to my comments and I am satisfied that you have satisfactorily addressed my concerns. You submitted a clean version of the revised manuscript. This version did not have any track changes so it was difficult to see exactly where in the text the changes had been made (although you included some of the changes in your responses to reviewers document).

Response:

Thank you for acknowledging our responses. Due to the large number of review comments for the first time, the content of the manuscript has undergone major changes. In order to present a complete and clear article idea, we did not mark the revised manuscript. (If marked, possibly all text content will be marked). I'm sorry that this behavior has caused you trouble, we will correct it in future work, thank you for pointing out this problem.

Reviewer #4:

1. Comment:

The authors addressed the concerns of the previous referee. However, the three reviewers raised the suspicious problem of LRP3 antibody western blot results, and the author solved the problem by changing and purchasing a new antibody and repeating the test, which itself increased the suspicious nature of the data. Considering the whole project, LRP3 runs through the whole process, including cell growth and protein degradation in animal cells. If the important role of LRP3 is to be explained, and if the antibody is replaced, the GAPDH band should also be replaced. That is, if the internal reference is replaced, other antibody bands will also be changed, and other genes cannot use the previous antibody bands. Besides, there are many picture errors and inconsistent data, resulting in doubts about the authenticity of the conclusion.

Response:

Thank you for your professional advice. We are sorry for the confusion and doubts caused by the previous reply. The first thing to explain is that we did not replace or change the LRP3 antibody,

but re-purchased the same LRP3 antibody with the same product number from the same company. This is because the amount of the previous antibody was not enough to complete the Western blot experiment, and the shelf life of the previous antibody had also expired.

Secondly, we did not just replace the unclear LRP3 bands. In order to meet the reviewer's request and ensure the authenticity and reliability of the experimental results, we have repeated all Western blots in this study. All bands showed in this edition of the figures are new. The uncropped and original full film figures with indicate molecular weight markers are provided in the supplementary data.

Finally, we not only re-purchased the LRP3 antibody, but in order to ensure the smooth progress of the experiment and the consistent production and storage time of all antibodies, we re-purchased all the antibodies in this study with the same product number from the same company, including the GAPDH antibody.

2. Comment:

Figure 1b, in group of "TNF-a+Cholesterol", the pathological staining samples were from different specimens, especially for the COLII staining. The surface of the cartilage in COLII staining were intact, while the cartilage in Safranin O staining and Toluidine blue were seriously damaged. This is unreasonable.

Response:

Thank you for your professional advice. Sorry for the misunderstanding caused by the Figure 1b, the COL II pictures we presented were indeed from the same experimental treatment group. In order to avoid misunderstandings, we have re-selected images close to the same slice location as you requested.

3. Comment:

Figure 3i and 3k, there are also cases where the pictures are not consistent. The mouse knee cartilage of 12m in Figure 3i were damaged, while the cartilage in Figure 3k were relatively normal, why are the samples different in the same period?

Response:

Thank you for your professional advice. Sorry for the misunderstanding caused by the Figure

3i and 3k. According to your requirements, we replaced the specimens of the same mouse with close sections. Due to manual reasons, not all paraffin serial sections are perfect and usable, and sections may also suffer from curling, damage, etc. during the staining process, resulting in possible differences in the details of the presented images. But it should be clarified that the pictures we selected were all from the same experimental treatment group which including multiple mice.

We are very grateful for your comments with a rigorous and scientific research attitude, and our research group will pay more attention to such issues in future research.

4. Comment:

Figure 4k, the LRP staining indicated that both nucleus and cytoplasm were stained, while in figure s3a, the LRP staining were in nucleus, what is the specific location of the LRP3 staining in the cell? The location and intensity of cell staining are different. Please confirm whether the picture is misplaced.

Response:

Thank you for your professional advice. LRP3 has never been reported before, our study found that LRP3 was expressed in both cytoplasm and nucleus. In response to your question, what needs to be explained is that cartilage tissue is a special terminally differentiated tissue, mainly composed of a small number of chondrocytes and a large amount of extracellular matrix. Chondrocytes are scattered in cartilage lacunae distributed in the extracellular matrix. In our picture, the positively stained puncta are actually chondrocytes in the cartilage lacunae. The entire chondrocytes were positively stained, including their cytoplasm and nuclei.

Secondly, Figure 4k is the mouse cartilage tissue, while Figure S3a is the rat cartilage tissue. There are some differences in the tissue morphology between the mouse and rat cartilage tissue, which causing your misunderstanding.

In addition, since the extracellular matrix of mouse cartilage is looser than that of rats, non-specific staining of mouse extracellular matrix may occur during LRP3 immunohistochemical staining. This is the inadequacy of our staining process and we will continue to be improved in future research.

5. Comment:

Figure 6i, according to significance mark on the left, Ras signaling pathway were marked blue. According to the label, there should be no statistical significance for this pathway.

Response:

Thanks for your professional question. Glad you raised this question, Figure 6i shows the pathway that might be activated after knocking down LRP3 in rat chondrocytes. At the beginning of the study, we did not know why LRP3 could play a role in cartilage degeneration, and then we found that after knocking down LRP3, the expression of SDC4 in cartilage was increased, which provided a direction for our research. But the new question at this time is how LRP3 regulates SDC4. Based on the KEGG results combined with previous literature, we found that the activation of the Ras/Raf/MEK/ERK signaling pathway is not only related to the occurrence and development of OA, but also closely related to the regulation of SDC4. Then we put forward the hypothesis that LRP3 upregulates SDC4 by activating the Ras pathway, and verified it with Western blot experiments, and got positive results. High-throughput RNA-sequencing provides direction for our research, but the results need to be validated, and there is a possibility of inconsistent results. Moreover, the results of KEGG are at the transcriptome level, but we use Western blot to verify at the protein level, which may be one of the reasons for the inconsistent results.

6. Comment:

Figure s1b, the staining for COL2A1 and LRP3 were overexposed, which can be seen from DAPI. In addition, the expression of LRP3 was inconsistent in the upper and lower pictures. The upper picture showed that it was expressed in the nucleus and the next one was not. It is suspected of manual brightness adjustment, resulting in unreliable data.

Response:

Thanks for your professional question. We're sorry to bother you with the results presented here. It should be explained that all the pictures were taken under the same conditions, and there was no behavior considered to adjust the brightness of the pictures in the control group. LRP3 is expressed in both nucleus and cytoplasm of chondrocytes. Our research group will learn from experience and present better image quality in future work. Thanks again for your opinion on our team.

7. Comment:

Figure s3b, LV silrp3 sham group safranin O staining showed cartilage surface were damaged, while LRP3 group showed intact cartilage surface and different cell number and arrangement of chondrocytes. The data were not accurate enough. Similarly, in the lv-silrp3 ACLT group, the cartilage samples of safranin O staining and the samples of LRP3 and col2 are not the same. The unreliable data resulted in the weakening of the authenticity of the conclusion.

Response:

Thanks for your professional question. Sorry for the misunderstanding caused by the Figure s1b. Similarly, the pictures we selected were indeed from the same experimental treatment group, since the same experimental treatment group contained multiple rats. In order to present a good staining picture, our specimens may have come from different rats in the same group without affecting the overall expression trend. And for the rigor and reliability of the experiment, we performed semi-quantitative analysis of immunohistochemical staining, including all samples from the same experimental group. In accordance with your request, we have modified the pictures and selected the same group of specimens from the same rat with close sections for presentation.

8. Comment:

Figure s6a, the LRP^{-/-} ACLT group cartilage safranin O staining was seriously damaged, while the cartilage surface in LRP3 group and col2 group was intact. The pictures were selected from different periods?

Response:

Thanks for your professional question. Sorry for the misunderstanding caused by the Figure s6a. Likewise, the images we selected before were all from the same experimental treatment group. In accordance with your request, we have modified the LRP^{-/-} ACLT group pictures and selected the same group of specimens from the same mouse with close sections for presentation. It needs to be pointed out and explained here is that severe cartilage destruction may cause edge effects during immunohistochemical staining, resulting in non-specific staining. In order to avoid this phenomenon affecting the presentation of the results, we chose the pictures with weaker edge effects in the same experimental treatment group before, which may misunderstand you, but the experimental results are not deliberately falsified.

9. Comment:

Figure s8b, it was obvious that the staining of col4-sv group was not the same as that of col4-sv group in the same cartilage sample. Please explain why the staining of col4-sv group was not the same as that of col4-sv group in the same sample.

Response:

Dear reviewer, it may be due to your typo, we are sorry that we did not understand your question, there is no col4-sv group in Figure S8b.

First of all, it is certain that the pictures in each of our groups are selected from the samples of the same experimental treatment group.

Second, SDC4 is only expressed in chondrocytes located in the cartilage lacuna and not in the extracellular matrix of cartilage.

Finally, COL2 is expressed both in chondrocytes and extracellular matrix of chondrocytes. When cartilage tissue degenerates, the expression of COL2 in the chondrocytes themselves first decreased, then the expression of COL2 secreted by chondrocytes into the extracellular matrix is reduced. In healthy cartilage tissue (Sham lv-con335 group), both chondrocytes and extracellular matrix highly expressed COL2, while when cartilage degenerated (Sham lv-Sdc4 group, ACLT lv-con335 group, ACLT lv-Sdc4 group), the expression of COL2 in chondrocytes was significantly decreased, which then affected the expression of COL2 in the extracellular matrix. Hope my answer satisfied you.

REVIEWERS' COMMENTS

Reviewer #4 (Remarks to the Author):

I have no more questions.

REVIEWERS' COMMENTS

Reviewer #4 (Remarks to the Author):

I have no more questions.

Response: Thank you for your approval of our work, and we will continue to make efforts to do better research.

Sincerely yours,

Yingfang Ao on behalf of the authors.